# FAST UNSUPERVISED GROUND METRIC LEARNING WITH TREE-WASSERSTEIN DISTANCE

**Kira M. Düsterwald[1,2,+], Samo Hromadka[1], and Makoto Yamada[2]**

[1] Gatsby Computational Neuroscience Unit, University College London, United Kingdom
[2] Machine Learning and Data Science Unit, Okinawa Institute of Science and Technology, Japan
[+] kira.dusterwald.21@ucl.ac.uk

## ABSTRACT

The performance of unsupervised methods such as clustering depends on the choice of distance metric between features, or ground metric. Commonly, ground metrics are decided with heuristics or learned via supervised algorithms. However, since many interesting datasets are unlabelled, unsupervised ground metric learning approaches have been introduced. One promising option employs Wasserstein singular vectors (WSVs), which emerge when computing optimal transport distances between features and samples simultaneously. WSVs are effective, but can be prohibitively computationally expensive in some applications: $\mathcal{O}(n^2 m^2 (n \log(n) + m \log(m)))$ for $n$ samples and $m$ features. In this work, we propose to augment the WSV method by embedding samples and features on trees, on which we compute the tree-Wasserstein distance (TWD). We demonstrate theoretically and empirically that the algorithm converges to a better approximation of the standard WSV approach than the best known alternatives, and does so with $\mathcal{O}(n^3 + m^3 + mn)$ complexity. In addition, we prove that the initial tree structure can be chosen flexibly, since tree geometry does not constrain the richness of the approximation up to the number of edge weights. This proof suggests a fast and recursive algorithm for computing the tree parameter basis set, which we find crucial to realising the efficiency gains at scale. Finally, we employ the tree-WSV algorithm to several single-cell RNA sequencing genomics datasets, demonstrating its scalability and utility for unsupervised cell-type clustering problems. These results poise unsupervised ground metric learning with TWD as a low-rank approximation of WSV with the potential for widespread application.

## 1 INTRODUCTION

The geometry of data structures can help to solve unsupervised learning tasks. For example, clustering methods such as $k$-means compare groupings based on a distance between samples informed by a distance on features. Inherently, the heuristic that one chooses for the distance matrix or function determines the outcome of the algorithm. So, how should one choose or learn this distance?

$k$-means gives equal weighting to the distance between each pair of features. From an optimal transport theory perspective, it could be argued that a more holistic sample distance is given by the Wasserstein distance: the minimum cost using an optimal mapping between samples, weighted by the cost, or *ground metric*, between their features. For example, samples might be documents, and features words. Then the distance/cost between two documents is determined by the relative expression of words in each, mappings between them, and the ground metric between words. Hence the problem of learning sample distances is determined by the choice of the (feature) ground metric.

State-of-the-art methods usually employ heuristics for the ground metric, commonly Euclidean, with embeddings for the features, e.g. word2vec for documents (Kusner et al., 2015) and gene2vec in bioinformatics (for identifying cell type from gene expression; Zou et al. (2019); Du et al. (2019)). However, embeddings are not always available, or may be difficult to learn. This arises, for instance, when doing single-cell RNA sequencing on a new tissue for which marker genes are not known.

We are interested in unsupervised ways to learn the ground metric. We do so by augmenting an iterative method by Huizing et al. (2022). By harnessing the geometrical properties inherent in

optimising distances between samples, which are expressed as vectors of features, Huizing et al. (2022) learn ground metrics via positive singular vectors, entirely unsupervised. Their algorithm uses power iterations to first learn distances on *samples* (expressed in histograms over features) and in turn learn distances on *features* (histograms over samples).

To illustrate the idea, consider the *words* "Sentosa" and "Singapore", which one might expect to be close in word-space. If the metric between *documents* in the corpus that contain both of these words is close, we learn a smaller distance in the word dictionary between "Sentosa" and "Singapore". This in turn decreases the distance between documents containing these words. Such an approach could be applied to gene expression (features) in cells (samples), V1 neuronal activity (features) corresponding to representation of images (samples), and other unlabelled high-dimensional data.

While effective, Huizing et al. (2022)'s algorithm is expensive to run as it requires multiple pairwise distance computations per iteration. Inspired by Yamada et al. (2022), we propose a method that reduces the algorithmic complexity of each iteration by at least a quadratic factor by representing the data on a tree and learning weights on tree edges, rather than the full pairwise distance matrix. Additionally, we prove a lemma that suggests a powerful, fast recursive algorithm to compute the set of basis vectors for the matrix of pairwise leaf paths on any tree. We provide source code at: `https://github.com/kiradust/tree-wsv/`.

The unsupervised tree-Wasserstein method for finding ground metrics offers a geometric low-rank approximation interpretation of the full unsupervised ground metric learning problem. We show that it performs well on a toy dataset, and scales favourably to large 5000-10000 single-cell RNA-sequencing datasets for cell type classification.

## 1.1 PREVIOUS AND RELATED WORK

In this section we introduce optimal transport distances, tree-Wasserstein distances, inverse optimal transport and unsupervised ground metric learning.

**Optimal transport:** Optimal transport (OT) can be thought of as a "natural geometry" for probability measures (Peyré & Cuturi, 2019). The classical OT problem – attributed to Monge (1781) – asks how to find an optimal transport plan between two probability distributions subject to some cost. This is not always guaranteed to have a solution, and has non-linear constraints, which makes finding solutions difficult. For this reason, the modern OT usage hinges on a definition by Kantorovich (1940), allowing *probabilistic* mappings.

Consider the empiric discrete OT problem. Let $\mu = \sum_i^n a_i \delta_{\boldsymbol{x_i}}$ and $\nu = \sum_j^n b_j \delta_{\boldsymbol{x_j}}$ be discrete probability distributions ("histograms", or normalised bin counts), with $\delta_{\boldsymbol{x_i}}$ the Dirac delta function at position $\boldsymbol{x_i}$. Then the OT problem is to find optimal $\boldsymbol{P}$ satisfying:

$$\mathcal{W}_{1,\boldsymbol{C}}(\mu,\nu) = \min_{\boldsymbol{P} \in \mathbb{R}_+^{n \times n}} \langle \boldsymbol{P}, \boldsymbol{C} \rangle = \min_{\boldsymbol{P} \in \mathbb{R}_+^{n \times n}} \sum_i \sum_j P_{i,j} C_{i,j} \tag{1}$$

where $\boldsymbol{P}\mathbf{1}_n = \boldsymbol{a}, \boldsymbol{P}^\top \mathbf{1}_n = \boldsymbol{b}$ and $\boldsymbol{C} \in \mathbb{R}_+^{n \times n}$ is some cost function. $\mathcal{W}_{1,\boldsymbol{C}}(\mu,\nu)$ is known as the (1)-Wasserstein distance. In this paper, we shall drop the 1 and assume that $\mathcal{W}_{\boldsymbol{C}}$ is induced by the pairwise distance matrix $\boldsymbol{C}$ as its ground metric.

For example, in document similarity, $\boldsymbol{x_i}, \boldsymbol{x_j}$ are words (or embedding vectors) and $\boldsymbol{a}$ (resp. $\boldsymbol{b}$) tells us the frequency of words in a given document $\mu$ (resp. $\nu$). Usually, $\boldsymbol{C}$ is assumed to be the Euclidean distance between word embeddings, and then $\mathcal{W}_{\boldsymbol{C}}(\mu,\nu)$ is the word mover's distance (Kusner et al., 2015). Here, we instead take the approach of *learning* $C$ in an efficient manner, rather than utilising pre-trained embeddings, for documents represented as bag-of-words.

**Computationally efficient OT:** Solving the OT problem to compute $\mathcal{W}_{\boldsymbol{C}}$ via linear programming has complexity $\mathcal{O}(n^3 \log n)$ (Peyré & Cuturi, 2019). Cuturi (2013) suggested using the celebrated Sinkhorn's algorithm, which speeds up the calculation through entropic-regularised OT to $\mathcal{O}(n^2)$. Sliced-Wasserstein distance (SWD), in which $\mathcal{W}_{\boldsymbol{C}}$ is computed via projection to a one-dimensional subspace, improves on this complexity to $\mathcal{O}(n \log n)$ (Kolouri et al., 2016). SWD can be expressed as a special case of a geometric embedding, tree-Wasserstein distance (TWD), when the tree structure comprises a root connected to leaves directly (Indyk & Thaper, 2003; Le et al., 2019). TWD has linear complexity, and so is faster even than SWD (Le et al., 2019).

**Tree-Wasserstein distance (TWD):** The tree-Wasserstein distance represents samples on a tree. Consider a tree $\mathcal{T}$ with tree metric $d_{\mathcal{T}}$ (the unique and shortest path between any two nodes on $\mathcal{T}$). Let $\boldsymbol{w} \in \mathbb{R}_+^{N-1}$ be the vector of weights between nodes and $\boldsymbol{Z} \in \{0,1\}^{(N-1) \times N_{\text{leaf}}}$ the tree parameter (where $N$ is the number of nodes in the tree and $N_{\text{leaf}}$ the number of leaves), defined by $Z_{i,j} = 1$ if the edge from a non-root node $i$ is from a parent of leaf $j$ or $i = j$, and 0 otherwise. Given a tree metric $d_{\mathcal{T}}$ on leaves $\boldsymbol{x}, \boldsymbol{y}$ and transport plan $\pi$ between measures $\mu, \nu$, the TWD can be written as $\mathcal{W}_{\mathcal{T}}(\mu, \nu) = \inf_{\pi \in U(\mu,\nu)} \int_{\mathcal{X} \times \mathcal{Y}} d_{\mathcal{T}}(\boldsymbol{x}, \boldsymbol{y}) d\pi(\boldsymbol{x}, \boldsymbol{y})$, where $U(\mu, \nu) = \{\pi \in \mathcal{M}_+^1(\mathcal{X} \times \mathcal{Y}) : P_{\mathcal{X}\#}\pi = \mu, P_{\mathcal{Y}\#}\pi = \nu\}$ is the set of joint probability distributions with marginals $\mu$ on $\mathcal{X}$ and $\nu$ on $\mathcal{Y}$ (Le et al., 2019; Yamada et al., 2022).

Furthermore, TWD takes on a closed analytical form and therefore can be computed in $\mathcal{O}(\dim(\boldsymbol{w}))$ (Takezawa et al., 2021):

$$\mathcal{W}_{\mathcal{T}}(\mu, \nu) = ||\text{diag}(\boldsymbol{w})\boldsymbol{Z}(\boldsymbol{x} - \boldsymbol{y})||_1 \tag{2}$$

TWD is a good empirical approximation of the usual 1-Wasserstein distance (Yamada et al., 2022).

**The inverse OT problem and ground metric learning:** Leveraging the geometric nature of the Wasserstein/OT distance for unsupervised learning tasks relies on having a good idea of the ground metric between features that is then "lifted" to the OT distance between samples. Heuristics for ground metrics can be useful, especially with embeddings (Kusner et al., 2015; Du et al., 2019), but we are interested in broader, principled ways to find ground metrics.

Inverse optimal transport provides one solution: given a transport plan $\boldsymbol{P}$, find the distance matrix $\boldsymbol{C}$. For matching problems based on recommendation systems, for example the marriage dataset where preferred pairings and the associated features (i.e. the transport plan) are known, inverse OT can find the underlying distance between features. While in general this problem is under-constrained, solutions exist given sufficient constraints on $\boldsymbol{C}$ (Paty & Cuturi, 2020; Li et al., 2019; Stuart & Wolfram, 2019). However, inverse OT requires access to the full transport plan. In practice and for high-dimensional features, such as gene-cell data, this is not feasible.

An alternative is to use *partial* information about distance or similarity to learn the ground metric, for example with supervised or semi-supervised learning (Cuturi & Avis, 2014). A related but distinct concept is the idea of supervised learning of *sample* Wasserstein distances (Huang et al., 2016) and supervised TWD (Takezawa et al., 2021); here, the (feature) ground metric is still assumed to be Euclidean, but information about distances is used to learn metrics between samples.

**Unsupervised ground metric learning with Wasserstein singular vectors (WSVs):** The fully unsupervised approach differs in that neither feature nor sample ground metric is assumed. Unsupervised techniques harness the relationship between the geometry of features (embedded in samples) and the geometry of samples (embedded in features) (Paty & Cuturi, 2020; Huizing et al., 2022).

Consider some data matrix $\boldsymbol{X} \in \mathbb{R}_+^{n \times m}$. Let $\boldsymbol{a_i} \in \mathbb{R}_+^m$ be a sample (normalised row such that it sums to 1) of $X$, and $\boldsymbol{b_k} \in \mathbb{R}_+^n$ be a feature (normalised column). These are just histograms "embedding" a sample as a probability distribution over features, and vice versa. For example, the $\boldsymbol{a_i}$ (of which there are $n$) could be a histogram of words in a document, and the $\boldsymbol{b_k}$ ($m$) documents containing a given word. As another example, $\boldsymbol{a_i}$ might represent cells (samples) as histograms over gene expression, where $\boldsymbol{b_k}$ are the genes (features) as histograms over cells that contain each gene.

Huizing et al. (2022) learn ground metric matrices $\boldsymbol{A} \in \mathbb{R}_+^{n \times n}$, $\boldsymbol{B} \in \mathbb{R}_+^{m \times m}$ satisfying the fixed point equations:

$$A_{i,j} = \frac{1}{\lambda_A}\mathcal{W}_{\boldsymbol{B}}(\boldsymbol{a_i}, \boldsymbol{a_j}), \qquad B_{k,l} = \frac{1}{\lambda_B}\mathcal{W}_{\boldsymbol{A}}(\boldsymbol{b_k}, \boldsymbol{b_l}), \tag{3}$$

$\forall k, l \in \{1, .., m\}, i, j \in \{1, ..., n\}$ and some $(\lambda_A, \lambda_B) \in \mathbb{R}_+^2$. Note that as opposed to Huizing et al. (2022), we use $\boldsymbol{A}$ to denote the ground metric on the $\boldsymbol{a_i}$ and $\boldsymbol{B}$ the ground metric on the $\boldsymbol{b_k}$. Power iterations are used to compute the ground metric matrices via repeated application of a mapping $\Phi$ that "lifts ground metrics to pairwise distances":

$$\Phi_{\boldsymbol{A}}(\boldsymbol{A})_{k,l} := W_{\boldsymbol{A}}(\boldsymbol{b_k}, \boldsymbol{b_l}) + \tau||\boldsymbol{A}||_\infty R(\boldsymbol{b_k} - \boldsymbol{b_l}),$$

where $R$ is a norm regulariser and $\tau > 0$. A symmetric definition holds for $\boldsymbol{B}$. Then the equivalent problem is to find Wasserstein singular vectors such that there exist $(\lambda_A, \lambda_B) \in (\mathbb{R}_+^*)^2$ satisfying $\Phi_{\boldsymbol{B}}(\boldsymbol{B}) = \lambda_A \boldsymbol{A}, \Phi_{\boldsymbol{A}}(\boldsymbol{A}) = \lambda_B \boldsymbol{B}$ (equal to (3) when $\tau = 0$). Maximal singular vectors can be extracted with power iterations: $\boldsymbol{A}_{t+1} = \frac{\Phi_{\boldsymbol{B}}(\boldsymbol{B}_t)}{||\Phi_{\boldsymbol{B}}(\boldsymbol{B}_t)||_\infty}, \qquad \boldsymbol{B}_{t+1} = \frac{\Phi_{\boldsymbol{A}}(\boldsymbol{A}_t)}{||\Phi_{\boldsymbol{A}}(\boldsymbol{A}_t)||_\infty}.$

The complexity of a single power iteration is $\mathcal{O}(n^2 m^2 (n \log(n) + m \log(m)))$, since it requires computing $m^2$ Wasserstein distances where each distance is $\mathcal{O}(n^3 \log(n))$, then $n^2$ Wasserstein distances where each distance is $\mathcal{O}(m^3 \log(m))$. Huizing et al. (2022) propose to reduce this complexity via stochastic optimisation (which suffers from requiring far more iterations to converge) and entropic regularisation, or "Sinkhorn singular vectors" (SSV), which improves the complexity to at best $\mathcal{O}(n^2 m^2)$. There are no other known reductions in complexity of this unsupervised method. Tong et al. (2022) propose an efficient $L_1$-embedding of the Wasserstein distance on graphs in the unbalanced OT regime; however, the underlying ground metric is still assumed geodesic.

While our approach to improve complexity through embedding samples and features as leaves on trees was inspired from the TWD literature, several authors have explored the relationship between samples (rows) and features (columns) as distributions over the other in general (Ankenman, 2014; Gavish & Coifman, 2012), including on graphs (Shahid et al., 2016) and in tree-embeddings (Ankenman, 2014; Mishne et al., 2018; Yair et al., 2017). These methods do not learn the ground metric in the same way as our proposal, but Mishne et al. (2018), Ankenman (2014) and Yair et al. (2017) describe iterative metric-learning of the tree metric and tree construction, which is related.

## 1.2 Contributions

We propose to improve on the WSV computational bottleneck by embedding the dataset in trees and approximating $\mathcal{W}_C$ with the TWD. We name our method Tree-WSV. Our findings position Tree-WSV as a fast, low-rank approximation of the standard WSV approach. We show that:

- One can learn a complete set of strictly positive tree edge weights by solving a non-singular system of linear equations (Algorithm 1), and hence solve for the entire Wasserstein distance matrix on features and samples, respectively.

- This system is determined by a pairwise leaf-to-leaf paths matrix whose rank is guaranteed to be equal to the number of edges in any tree with root degree $\geq 3$. This path matrix's basis set can be found via a recursive algorithm, allowing fast computation of edge weights.

- The Tree-WSV approach is a quadratic order more computationally efficient than WSV and outperforms Sinkhorn entropy regularisation (SSV) in terms of accuracy on toy data.

- Tree-WSV, combined with meta-iterations, scales to single-cell RNA sequencing datasets of size over $5000 \times 5000$, on which it is also faster and at least as accurate as SSV.

## 2 Unsupervised tree-Wasserstein ground metric learning

### 2.1 TWD singular vectors approximate standard Wasserstein SVs

Let $\boldsymbol{A} = \{\boldsymbol{a_i}\}, \boldsymbol{B} = \{\boldsymbol{b_k}\}, i \in \{1, ..., n\}, k \in \{1, ..., m\}$ be the set of normalised rows (samples) and set of normalised columns (features) respectively of the data matrix $\boldsymbol{X} \in \mathbb{R}_+^{n \times m}$. We embed $\boldsymbol{A}$ and $\boldsymbol{B}$ in respective trees $\mathcal{T}_{\boldsymbol{A}}$ and $\mathcal{T}_{\boldsymbol{B}}$, such that $\{\boldsymbol{a_1}, ..., \boldsymbol{a_n}\}$ are the leaves of $\mathcal{T}_{\boldsymbol{A}}$ and $\{\boldsymbol{b_1}, ..., \boldsymbol{b_m}\}$ the leaves of $\mathcal{T}_{\boldsymbol{B}}$, as illustrated in Fig. 1. Let $\boldsymbol{Z^{(A)}} \in \{0, 1\}^{(N-1) \times n}$, $\boldsymbol{Z^{(B)}} \in \{0, 1\}^{(M-1) \times m}$ be the tree parameters of each tree, where $N$ and $M$ are the number of nodes in $\mathcal{T}_{\boldsymbol{A}}$ and $\mathcal{T}_{\boldsymbol{B}}$ respectively. Let $\boldsymbol{w_A} \in \mathbb{R}_+^{N-1}, \boldsymbol{w_B} \in \mathbb{R}_+^{M-1}$ be the vectors of edge weights. We can port the WSV equations (3) to this tree setting:

**Proposition 2.1.** *The WSV fixed point equations* (3) *can be expressed on the trees* $\mathcal{T}_A, \mathcal{T}_B$ *as:*

$$d_{\mathcal{T}_{\boldsymbol{A}}}(\boldsymbol{a_i}, \boldsymbol{a_j}) = \frac{1}{\lambda_A} \mathcal{W}_{\mathcal{T}_{\boldsymbol{B}}}(\boldsymbol{a_i}, \boldsymbol{a_j}), \qquad d_{\mathcal{T}_{\boldsymbol{B}}}(\boldsymbol{b_k}, \boldsymbol{b_l}) = \frac{1}{\lambda_B} \mathcal{W}_{\mathcal{T}_{\boldsymbol{A}}}(\boldsymbol{b_k}, \boldsymbol{b_l}),$$

*where the singular vector update is to find* $\boldsymbol{w_A}$ *(and symmetrically* $\boldsymbol{w_B}$*) such that* $\forall i, j,$

$$\lambda_A \boldsymbol{w_A}^\top \left( \boldsymbol{z_i^{(A)}} + \boldsymbol{z_j^{(A)}} - 2\boldsymbol{z_i^{(A)}} \circ \boldsymbol{z_j^{(A)}} \right) = \left\| \mathrm{diag}(\boldsymbol{w_B}) \boldsymbol{Z^{(B)}} (\boldsymbol{a_i} - \boldsymbol{a_j}) \right\|_1, \qquad (4)$$

*where* $\boldsymbol{z_i^{(A)}}$ *is the ith column of* $\boldsymbol{Z^{(A)}}$, $\circ$ *denotes element-wise product and* $(\lambda_A, \lambda_B) \in (\mathbb{R}_+^*)^2$.

**Proof:** *Appendix A.* □

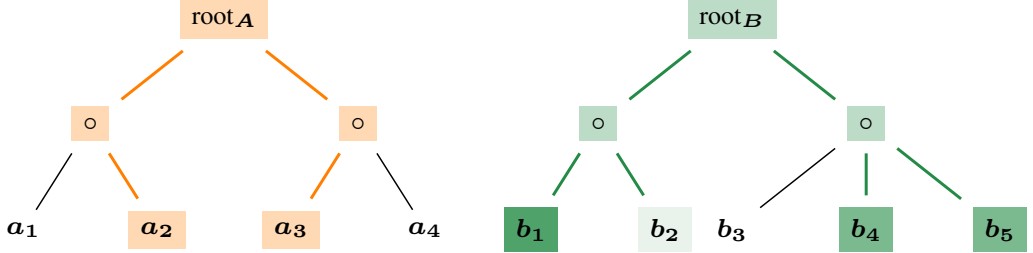

Figure 1: Tree embeddings for samples $\boldsymbol{A}$ as leaves in $\mathcal{T}_{\boldsymbol{A}}$ (left) and features $\boldsymbol{B}$ as leaves in $\mathcal{T}_{\boldsymbol{B}}$ (right). The tree metric $d_{\mathcal{T}_{\boldsymbol{A}}}(\boldsymbol{a_2}, \boldsymbol{a_3})$ is shown as the shortest path between these leaves in orange on the left. Equivalently, we can use the relative expression of the features $\{\boldsymbol{b_1}, \boldsymbol{b_2}, \boldsymbol{b_3}, \boldsymbol{b_4}, \boldsymbol{b_5}\}$ that represent $\boldsymbol{a_2}, \boldsymbol{a_3}$ respectively (as shown in different hues of green on the right) to compute a TWD, $\mathcal{W}_{\mathcal{T}_{\boldsymbol{B}}}(\boldsymbol{a_2}, \boldsymbol{a_3})$, in $\mathcal{T}_{\boldsymbol{B}}$. We assume $d_{\mathcal{T}_{\boldsymbol{A}}}(\boldsymbol{a_2}, \boldsymbol{a_3})$ is equal to $\mathcal{W}_{\mathcal{T}_{\boldsymbol{B}}}(\boldsymbol{a_2}, \boldsymbol{a_3})$ to learn a good embedding.

For intuition about Proposition 2.1, choose two samples $\boldsymbol{a_i}, \boldsymbol{a_j}$ that are leaves in one of the trees; $\mathcal{T}_{\boldsymbol{A}}$ (Fig. 1). The tree metric $d_{\mathcal{T}_{\boldsymbol{A}}}$ is the shortest path between these leaves with edges in $\mathcal{T}_{\boldsymbol{A}}$. $\boldsymbol{a_i}, \boldsymbol{a_j}$ can also be expressed as histograms over features which are leaves on $\mathcal{T}_{\boldsymbol{B}}$, i.e. $\boldsymbol{a_i} = \sum_{k=1}^m c_{\boldsymbol{b_k}}^{(\boldsymbol{a_i})} \boldsymbol{b_k}$, where $c_{\boldsymbol{b_k}}^{(\boldsymbol{a_i})} \geq 0$ is the relative expression of each $\boldsymbol{b_k}$ in the histogram, so that $\sum_{k=1}^m c_{\boldsymbol{b_k}}^{(\boldsymbol{a_i})} = 1$. The 1-Wasserstein distance $\mathcal{W}_{\boldsymbol{B}}(\boldsymbol{a_i}, \boldsymbol{a_j})$ between the two samples uses some unknown cost matrix $\boldsymbol{C}$ so that $\mathcal{W}_{\boldsymbol{B}}(\boldsymbol{a_i}, \boldsymbol{a_j}) = \min_{\boldsymbol{P}} \sum_k \sum_l P_{k,l} C_{k,l}$ where $\boldsymbol{P} \mathbf{1}_n = \boldsymbol{a_i}$ and $\boldsymbol{P}^\top \mathbf{1}_n = \boldsymbol{a_j}$. From Yamada et al. (2022), $\mathcal{W}_{\boldsymbol{B}}(\boldsymbol{a_i}, \boldsymbol{a_j})$ is approximated by the TWD between $\boldsymbol{a_i}, \boldsymbol{a_j}$ on $\mathcal{T}_{\boldsymbol{B}}$, which we can express through summing proportions of $d_{\mathcal{T}_{\boldsymbol{B}}}$ between the leaves in $\mathcal{T}_{\boldsymbol{B}}$: $d_{\mathcal{T}_{\boldsymbol{A}}}(\boldsymbol{a_2}, \boldsymbol{a_3}) = \mathcal{W}_{\mathcal{T}_{\boldsymbol{B}}}(\boldsymbol{a_2}, \boldsymbol{a_3})$.

We can utilise Proposition 2.1 to learn low-rank approximations of $\mathcal{W}_{\mathcal{T}_{\boldsymbol{B}}}(\boldsymbol{a_i}, \boldsymbol{a_j})$ and $\mathcal{W}_{\mathcal{T}_{\boldsymbol{A}}}(\boldsymbol{b_k}, \boldsymbol{b_l})$. Let $\boldsymbol{y_{i,j}} := \boldsymbol{z_i}^{(A)} + \boldsymbol{z_j}^{(A)} - 2\boldsymbol{z_i}^{(A)} \circ \boldsymbol{z_j}^{(A)} = \mathrm{XOR}\{\boldsymbol{z_i}^{(A)}, \boldsymbol{z_j}^{(A)}\} \in \{0,1\}^{N-1}$ and concatenate the $\boldsymbol{y_{i,j}}$ into a tensor $\mathbf{Y}^{(A)} \in \{0,1\}^{(N-1) \times n \times n}$ (i.e. all distinct nodes in the paths of all sets of pairwise leaves). Let $\boldsymbol{Y'} \in \{0,1\}^{(N-1) \times n^2}$ be the matrix created when the tensor $\mathbf{Y}$ is unravelled along $n \times n$ leaf-pairs. Equation 4 for all $i, j$ pairs then rewrites as a system of $n^2$ linear equations:

$$\lambda_A \boldsymbol{w_A}^\top \boldsymbol{Y'} = \mathcal{W}_{\mathcal{T}_{\boldsymbol{B}}} \text{, where } \mathcal{W}_{\mathcal{T}_{\boldsymbol{B}}} = \left\{ \left\| \mathrm{diag}(\boldsymbol{w_B}) \boldsymbol{Z}^{(B)} (\boldsymbol{a_i} - \boldsymbol{a_j}) \right\|_1, \forall i, j \right\} \in \mathbb{R}_+^{n^2}. \quad (5)$$

A symmetric system of equations holds with the roles of $\boldsymbol{w_A}, \boldsymbol{w_B}$ reversed.

## 2.2 NON-ZERO AND UNIQUE SOLUTIONS TO THE TREE WSV PROBLEM EXIST

Since Equation 5 is still a singular vector problem, it can be solved with power iterations. As in Huizing et al. (2022), we can show that a solution to the singular vector problem exists:

**Lemma 2.2.** *The singular value problem 5 has a solution* $(\boldsymbol{w}_A, \boldsymbol{w}_B)$.

**Proof:** *Appendix B* □

While Lemma 2.2 states that a solution exists, convergence to that solution is not guaranteed. However, in practice we always observed convergence, with dynamics that were consistent with the linear convergence rate reported in Huizing et al. (2022). We show empirical convergence for power iterations on different dataset sizes in Appendix E, Figure 4.

Each iteration of the singular value problem involves solving a non-negative least squares (NNLS) problem. In Theorem 2.4, contingent on Lemma 2.3, we show that as long as we restrict ourselves to trees whose root node has degree greater than 2 (which we can do, since ClusterTree's hyperparameters allow setting the number of children), NNLS finds unique, non-zero solutions to (5).

**Lemma 2.3.** *Let the number of nodes in a tree be* $N$, *including the root and* $n$ *leaves. Then there can be at most one non-leaf node of degree 2. In addition, if such a node exists, it is the root and* $\boldsymbol{Y'}$ *has rank* $N - 2$. *Otherwise, every degree of the tree is at least 3, and* $\boldsymbol{Y'}$ *has rank* $N - 1$.

**Proof:** *Appendix C* □

The proof of Lemma 2.3 suggests an efficient recursive algorithm that can also be used to find the basis set $\boldsymbol{U}$ for $\boldsymbol{Y'}$, and is detailed in Appendix D.

---

**Algorithm 1** Unsupervised ground metric learning with trees (Tree-WSV)

---

**Input:** dataset $\boldsymbol{X}$, size $n \times m$
**do once**
$\boldsymbol{A}, \boldsymbol{B} \leftarrow \{\boldsymbol{a_i} = \boldsymbol{X}_{row}^{(i)} / \sum \boldsymbol{X}_{row}^{(i)}\}, \{\boldsymbol{b_k} = \boldsymbol{X}_{col}^{(k)} / \sum \boldsymbol{X}_{col}^{(k)}\}$
$\boldsymbol{Z}^{(\boldsymbol{B})}, \boldsymbol{Z}^{(\boldsymbol{A})} \leftarrow \texttt{ClusterTree}(\boldsymbol{B}), \texttt{ClusterTree}(\boldsymbol{A})$
**if** $n$ and $m$ small (less than 500) **then**
    $\boldsymbol{Y}^{(\boldsymbol{A})}, \boldsymbol{Y}^{(\boldsymbol{B})} \leftarrow$ tensor $\{y_{i,j} = \text{XOR}\{\boldsymbol{z}_i^{(\boldsymbol{A})}, \boldsymbol{z}_j^{(\boldsymbol{A})}\}\}$, tensor $\{y_{k,l} = \text{XOR}\{\boldsymbol{z}_i^{(\boldsymbol{B})}, \boldsymbol{z}_j^{(\boldsymbol{B})}\}\}$
    $\boldsymbol{U}^{(\boldsymbol{A})}, \boldsymbol{U}^{(\boldsymbol{B})} \leftarrow$ basis sparse SVD$\big[\text{upper triangular}(\boldsymbol{Y}^{(\boldsymbol{A})}))\big], \big[\text{upper triangular}(\boldsymbol{Y}^{(\boldsymbol{B})}))\big]$
    $\boldsymbol{Z}_{\text{diff}}^{(\boldsymbol{A}_{k,l})}, \boldsymbol{Z}_{\text{diff}}^{(\boldsymbol{B}_{i,j})} \leftarrow \boldsymbol{Z}_{\boldsymbol{U}}^{(\boldsymbol{A})}(\boldsymbol{b_k} - \boldsymbol{b_l}) \, \forall \{\boldsymbol{b_k}, \boldsymbol{b_l}\} \in B_U, \; \boldsymbol{Z}_{\boldsymbol{U}}^{(\boldsymbol{B})}(\boldsymbol{a_i} - \boldsymbol{a_j}) \, \forall \{\boldsymbol{a_i}, \boldsymbol{a_j}\} \in A_U$
**else**
    compute $\boldsymbol{U}$s and $\boldsymbol{Z}_{\text{diff}}$s with recursive basis set algorithm (Appendix D)
**end if**
**initialise**
$\boldsymbol{w_A}, \boldsymbol{w_B} \leftarrow$ random vectors of length $N - 1, M - 1$, i.e. the number of edges in each tree
**repeat**
    **for** $t = 1$ **to** num iterations **do**
        $\mathcal{W}_{\mathcal{T}}(\boldsymbol{A}), \mathcal{W}_{\mathcal{T}}(\boldsymbol{B}) \leftarrow \{\big\|\text{diag}(\boldsymbol{w_A})\boldsymbol{Z}_{\text{diff}}^{(\boldsymbol{A}_{k,l})}\big\|_1, \forall k, l\}, \{\big\|\text{diag}(\boldsymbol{w_B})\boldsymbol{Z}_{\text{diff}}^{(\boldsymbol{B}_{i,j})}\big\|_1, \forall i, j\}$
        $\mathcal{W}_{\mathcal{T}}(\boldsymbol{A})_{\text{norm}}, \mathcal{W}_{\mathcal{T}}(\boldsymbol{B})_{\text{norm}} \leftarrow \mathcal{W}_{\mathcal{T}}(\boldsymbol{A}) / \|\mathcal{W}_{\mathcal{T}}(\boldsymbol{A})\|_\infty, \mathcal{W}_{\mathcal{T}}(\boldsymbol{B}) / \|\mathcal{W}_{\mathcal{T}}(\boldsymbol{B})\|_\infty$
        $\boldsymbol{w_A} \leftarrow \texttt{NNLS}\big[\boldsymbol{w_A}^\top \boldsymbol{U}^{(\boldsymbol{A})} = \mathcal{W}_{\mathcal{T}}(\boldsymbol{B})_{\text{norm}}\big]$
        $\boldsymbol{w_B} \leftarrow \texttt{NNLS}\big[\boldsymbol{w_B}^\top \boldsymbol{U}^{(\boldsymbol{B})} = \mathcal{W}_{\mathcal{T}}(\boldsymbol{A})_{\text{norm}}\big]$
    **end for**
**until** convergence: $\max(\|\boldsymbol{w_A} - \boldsymbol{w_A}(\text{previous})\|_1/n, \|\boldsymbol{w_B} - \boldsymbol{w_B}(\text{previous})\|_1/m) < \epsilon = 10^{-6}$

---

**Theorem 2.4.** *Given any tree $\mathcal{T}_A$ with leaves $\boldsymbol{A} = \{\boldsymbol{a_1}, ..., \boldsymbol{a_n}\}$ the rows of a data matrix $\boldsymbol{X} \in \mathbb{R}_+^{n \times m}$ such that the root node of $\mathcal{T}_A$ has degree 3 or more, and a tree $\mathcal{T}_B$ with leaves given by $\boldsymbol{B} = \{\boldsymbol{b_1}, ..., \boldsymbol{b_m}\}$ the columns of $\boldsymbol{X}$, there exists a unique non-negative solution for $\boldsymbol{w_A}$ in (5). Moreover, we can assume without loss of generality that the solution is strictly positive.*

**Proof:** *As the matrix $\boldsymbol{Y}'$ has rank $N - 1$, the solution $\boldsymbol{w_A}$ to $\lambda_A \boldsymbol{w_A}^\top \boldsymbol{Y}' = \mathcal{W}_{\mathcal{T}_B}$ is equivalent to the solution of $\lambda_A \boldsymbol{w_A}^\top \boldsymbol{Y}_{\boldsymbol{U}}' = (\mathcal{W}_{\mathcal{T}_B})_{\boldsymbol{U}}$, where $\boldsymbol{U}$ is a basis set of linearly independent columns of $\boldsymbol{Y}'$. $\boldsymbol{Y}_{\boldsymbol{U}}'$ is an invertible $(N - 1) \times (N - 1)$ matrix. Therefore, the NNLS problem*

$$\arg \min_{\boldsymbol{w_A}} \|\lambda_A \boldsymbol{w_A}^\top \boldsymbol{Y}_{\boldsymbol{U}}' - (\mathcal{W}_{\mathcal{T}_B})_{\boldsymbol{U}}\|_2^2 \quad \text{s.t. } \boldsymbol{w_A} \geq 0$$

*is a minimisation of a strictly convex function over a convex set, and has a unique solution (existence also follows directly from the Rouché–Capelli Theorem). Further, as in Yamada et al. (2022), if $\boldsymbol{w_A}$ contains any zeros, these can be removed, which corresponds to merging nodes of the tree $\mathcal{T}_A$.* $\square$

## 2.3 COMPUTATIONAL COMPLEXITY AND SPEED-UPS

Note that the sparse LASSO-based algorithm for learning $\boldsymbol{w}$ ("cTWD") proposed in Yamada et al. (2022) could also solve Equation 5. However, cTWD cannot be applied directly in our setting due to its computational complexity: solving each TWD-LASSO problem is $\mathcal{O}(M^3 + m^2 M^2)$, which is worse than the cubic complexity for NNLS (Efron et al., 2004), and cTWD requires computing all $n^2$ or $m^2$ TWDs at each iteration. Further, memory consumption from storing the full $\boldsymbol{Y}'$ matrix is quickly prohibitive.

Our Tree-WSV method suggests an efficient way of computing tree-Wasserstein singular vectors in Algorithm 1. Each inner power iteration computes tree-Wasserstein distances in time complexity $\mathcal{O}(n)$ rather than $\mathcal{O}(n^3 \log(n))$ for the 1-Wasserstein distance. Further, in each iteration we do not need to compute all $m^2$ distances, but are capped to $M < 2m - 1$, the length of the weight vector $\boldsymbol{w}$ (assuming no redundant nodes). There are a few additional speed-ups possible, detailed below.

First, note that we only need to construct trees once (without edge weights), using ClusterTree with the number of children $k$ set as $k > 2$ (Le et al. (2019)). Note that here, ClusterTree is a modification of QuadTree (Samet (1984) and extended to higher dimensions via a grid construction in Indyk & Thaper (2003)). Our method follows the implementation in Le et al. (2019) and Gonzalez

(1985). ClusterTree implemented in this way has complexity $\mathcal{O}(n\kappa)$ where $\kappa$ is the number of leaf-clusters (in our case set automatically by the hyperparameter controlling the depth of the tree) (Gonzalez, 1985). We compute $\boldsymbol{Z}^{(A)}, \boldsymbol{Z}^{(B)}$ once for each of $\boldsymbol{A}, \boldsymbol{B}$, and $\boldsymbol{Z}^{(A)}(\boldsymbol{b_k} - \boldsymbol{b_l})\ \forall k, l$ and $\boldsymbol{Z}^{(B)}(\boldsymbol{a_i} - \boldsymbol{a_j})\ \forall i, j$ just once.

Similarly, we only learn the $\boldsymbol{Y}$ tensors once with the tree structure. In fact, we do not need to learn the whole tensor: we just need a full-rank sub-matrix of $\boldsymbol{Y'}$ of rank $N - 1$, from Theorem 2.4, which we call $\boldsymbol{U}$ – the basis vector set for the matrix of all pairwise paths between leaves.

### 2.3.1 LEARNING THE BASIS VECTOR SET FOR ALL PAIRWISE LEAF-LEAF PATHS

There are two options to learn the basis set $\boldsymbol{U}$. Since $\boldsymbol{Y'}$ is sparse, for small dataset sizes, one can employ `scipy` sparse methods to find the basis set. Sparse QR decomposition or sparse singular value decomposition (SVD) can be used; in practice sparse SVD performed slightly faster.

Because this operation requires computing a large tensor $\boldsymbol{Y}$ reshaped into a long rectangular matrix of size $(N - 1) \times n^2$, the sparse method does not scale well with large $n, m$ from a memory-consumption point of view. In this case, we instead created a recursive algorithm as suggested from the constructive proof of Lemma 2.3. This method directly computes a basis set from just the $(N - 1) \times n$ (nodes by leaves) tree parameter matrix $\boldsymbol{Z}$. Recursions are done on at most the number of nodes less the number of leaves ($N - n < n - 1$ recursive loops); therefore, it is very fast and resists stack overflow. Stochasticity can be added by allowing randomness in path assignments, but since the result is still a basis set, this has minimal effect on finding the solution to the linear system of equations. The full recursive function is detailed in Appendix D. In practice, both approaches produced well-conditioned $\boldsymbol{U}$ basis set matrices (Section 3.2).

### 2.3.2 ITERATIVELY UPDATING THE WEIGHT VECTORS

At each iteration, we learn the vector $\boldsymbol{w_A}$ satisfying equation 5, using the basis set $\boldsymbol{U}^{(A)}$ of the coefficient matrix $\boldsymbol{Y'}$:

$$\lambda_A \boldsymbol{w_A}^\top \boldsymbol{U}^{(A)} = \mathcal{W}_{\mathcal{T}_B} \ , \text{ where } \ \mathcal{W}_{\mathcal{T}_B} = \left\{ \left\| \operatorname{diag}(\boldsymbol{w_B}) \boldsymbol{Z}_U^{(B)}(\boldsymbol{a_i} - \boldsymbol{a_j}) \right\|_1 \forall \boldsymbol{a_i}, \boldsymbol{a_j} \in \boldsymbol{A_U} \right\}.$$

Here, $\boldsymbol{U}^{(A)} \in \mathbb{R}^{(N-1) \times (N-1)}$ is the basis vector set for $\boldsymbol{Y'}$ which corresponds with pairs of vectors $\{\boldsymbol{a_i}, \boldsymbol{a_j}\} \in \boldsymbol{A_U}$, and $\boldsymbol{Z}_U^{(B)}$ is the sub-matrix of the $\boldsymbol{Z}$ matrix on these pairs $\{\boldsymbol{a_i}, \boldsymbol{a_j}\} \in \boldsymbol{A_U}$. We then symmetrically learn $\boldsymbol{w_B}$, and iterate.

From Theorem 2.4, a solution exists, and it can be found with a linear systems solver such as non-negative least squares (NNLS). NNLS is efficient, with cubic complexity on the size of $\boldsymbol{w_A}$ (i.e. $\mathcal{O}(N^3)$ and $\mathcal{O}(M^3)$ for $\boldsymbol{w_A}, \boldsymbol{w_B}$ respectively). Each compute of a pairwise TWD is linear in the length of the vectors ($m$ or $n$ for $\boldsymbol{a_i}, \boldsymbol{b_k}$ respectively), and there are $N - 1$ or $M - 1$ of these to compute. This gives overall complexity per power iteration: $\mathcal{O}\left(N^3 + mN + M^3 + nM\right) < \mathcal{O}\left(n^3 + m^3 + mn\right)$, using $N < 2n - 1, M < 2m - 1$. This is at least a quadratic order faster than $\mathcal{O}(n^2 m^2(n\log(n) + m\log(m)))$, the complexity for a WSV power iteration, and if $n \approx m$, an order faster than $\mathcal{O}\left(2n^2 m^2\right)$, the complexity for a SSV power iteration.

Overall, we achieve a significant theoretical gain in time complexity. Computationally efficient unsupervised ground metric learning with TWD (Tree-WSV) is summarised in Algorithm 1.

## 3 EXPERIMENTAL RESULTS ON TOY DATASET

We demonstrate the improved speed and accuracy of our method as compared to SSV on a synthetic dataset $\boldsymbol{X} \in \mathbb{R}^{n \times m}$, $n = 80, m = 60$, modified from prior work (Huizing et al., 2022). On various synthetic dataset sizes and ratios, we observed empiric convergence of power iterations (Appendix E) and confirmed that both the SVD and recursive basis set approaches are well-conditioned.

### 3.1 DATASET CONSTRUCTION AND EXPERIMENTAL METHODS

The dataset uses various translations of a unimodal periodic function on a 1D torus (with boundary conditions), such that $\boldsymbol{X}_{ik} = \exp\left(-\left(\frac{i}{n} - \frac{k}{m}\right)^2 / \sigma^2\right)$. On $\boldsymbol{X}$, the underlying (and learned) ground

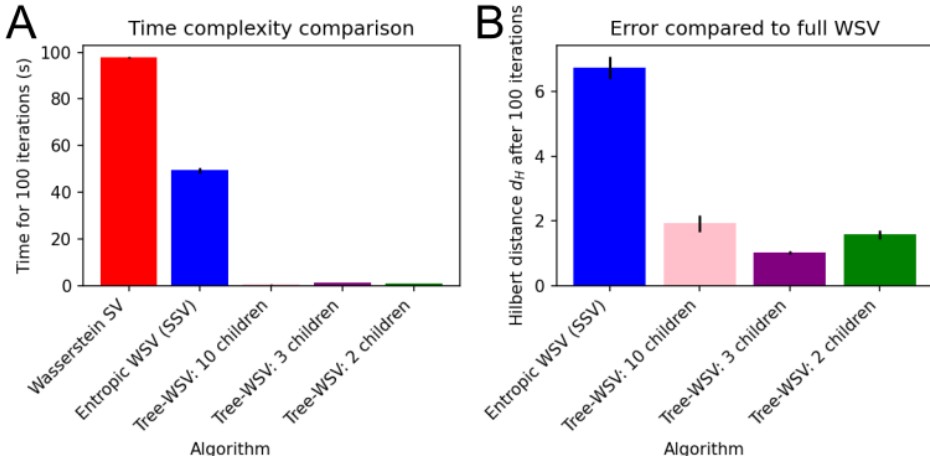

Figure 2: Comparison of (**A**) mean time complexity and (**B**) mean Hilbert distance between learned and WSV ground metric matrices recovered by different algorithms (lower is better). Tree-WSV algorithms are specified by their hyperparameter on the minimum number of children per node in the tree; all trees have maximum depth 5. Standard error of the mean is shown with black bars.

metrics look like $\left|\sin(\frac{i}{n} - \frac{j}{n})\right|$ (and symmetrically for $m$). We ran experiments 3 times for 100 iterations each on a CPU with different methods, using a translation of the periodic function, to which we added a 50% translation of the torus with half the magnitude. We set $\sigma = 0.01$. Thus $\boldsymbol{X}_{ik} \propto \exp\left(-\left(\frac{i}{n} - \frac{k}{m}\right)^2/\sigma^2\right) + 0.5\left(-\left(\frac{i}{n} - \frac{k}{m} + 0.5\right)^2/\sigma^2\right)$. In order to control against potential gains from tree construction based on preferred data ordering, we randomly permuted the dataset rows and columns before each experiment. We compare performance via computation time and the Hilbert metric between the learned cost and that recovered by the standard WSV algorithm (Fig. 2).

## 3.2 RESULTS

All tree-based methods were faster than both standard WSV and entropic-regularised WSV by 2-3 orders of magnitude (3-children trees took ∼1% of the time of the standard WSV, for example), as predicted by our theoretical time complexity calculations (Fig. 2 A). Interestingly, the tree-based methods had lower Hilbert distance error when compared to entropic WSV (Fig. 2 B). Tree methods were noted to converge in a similar amount of time (less than 10 iterations) to the standard WSV.

As expected from Lemma 2.3, the trees initialised with 10 children had lower $\boldsymbol{U}$-ranks than both 2- and 3-trees, and 3-trees had lower ranks than 2-trees. In general, trees with fewer children per node (higher rank) computed distance matrices slower, but produced more accurate results (Fig. 2). One exception follows directly from Lemma 2.3: 2-trees have rank $N - 2$ while 3-trees are rank $N - 1$, so the learned edge weight vector $\boldsymbol{w}$ is not guaranteed to be unique/optimal. As a result, the 2-trees performed worse in terms of accuracy than the 3-trees (Fig. 2 B).

We found that the basis set matrices $\boldsymbol{U}$ were well-conditioned for both the SVD and recursive approaches. For the SVD approach, the condition numbers were within $1 \pm 10^{-15}$ on dataset sizes 80x60, 100x200 and 500x500. For the recursive algorithm, the condition numbers were higher, 1000-3000 for random sample sizes between 1000 and 2000. Based on numerical precision, these higher condition numbers should not affect the accuracy of $\boldsymbol{w}$ to within 2 orders of magnitude.

## 4 EXPERIMENTAL RESULTS ON LARGE GENOMICS DATASETS

Single-cell RNA sequencing (scRNA-seq) is a powerful biotechnology method that measures gene expression in individual cells (Tang et al., 2009). scRNA-seq data can be used to determine cell types through clustering, useful for many applications. Clustering reliability across methods is limited by the choice of parameters and uncertainty in the number of clusters (Krzak et al., 2019). Traditional

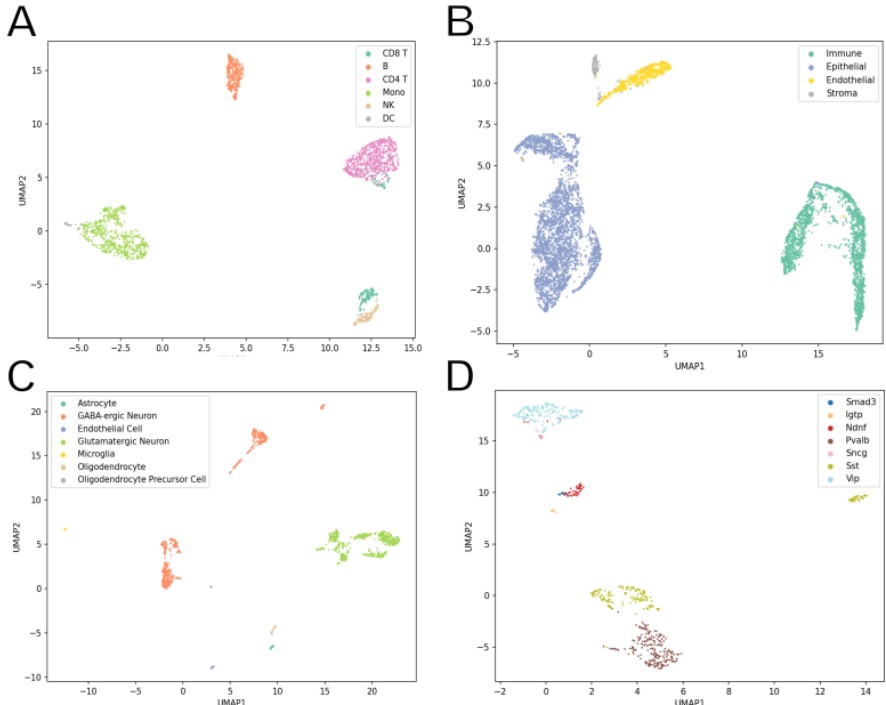

Figure 3: UMAP embeddings based on the learned tree-singular vectors for cells, and coloured by provided annotation: (**A**) PBMCs (Wolf et al., 2018); (**B**) Lung tissue (Sikkema et al., 2023); (**C**) Mouse V1 neural tissue, by broad types (Tasic et al., 2018); and (**D**) GABAergic neurons in the same mouse V1 neural tissue set as **C**, by subtype (Tasic et al., 2018).

methods assume a Euclidean metric on PCA embeddings, or use neural network model embeddings (Stuart et al., 2019; Wolf et al., 2018; Gayoso et al., 2022).

More recently, the gene mover distance has been used (Bellazzi et al., 2021; Du et al., 2019). Huizing et al. (2022) showed that Sinkhorn singular vectors can achieve state-of-the-art clustering scores on a small scRNA-seq dataset. We illustrate the potential utility of Tree-WSV by using it on the same dataset and achieving similar performance, as well as employing it on other datasets with improved performance. Finally, we showcase Tree-WSV on a large dataset prohibitive to the standard WSV approach.

## 4.1 CELL-TYPE CLUSTERING USING GENOMICS DATA

We first compare performance on the "PBMC 3k" dataset from 10x Genomics (Wolf et al., 2018), which consists of 2043 peripheral blood mononuclear cells with 6 broad types and 1030 genes. We also ran Tree-WSV on "Neurons V1", a dataset of scRNA-seq neuronal visual area 1 (V1) cells in mice by Tasic et al. (2018), consisting of 1468 cells with 7 broad types and 1000 genes, and a subset of the "Human Lung Cell Atlas (HLCA)" scRNA-seq consortium dataset of human respiratory tissue (Sikkema et al., 2023), consisting of 7000 cells with 4 broad types and 3923 genes. Details on preprocessing, annotation and experimental set-up are identical to Huizing et al. (2022) for PBMC and similar in the other cases, and provided in Appendix F. Experiments were run on a NVIDIA A100 Tensor Core GPU with JAX (Bradbury et al., 2018).

The average silhouette width (ASW) metric was used to compare performance on the clustering task. ASW measures how close each data point is to its own cluster compared to other clusters, using an input distance metric. Scores range between -1 and 1, with 1 being better.

## 4.2 RESULTS USING TREE-WSV META-ITERATIONS

We noted that silhouette scores for one iteration of the Tree-WSV algorithm – while quick to run – were low across datasets (0.07, 0.15 and 0.05 respectively). This is because the initial tree structure is calculated via ClusterTree, which uses Euclidean distances on the input to cluster nodes based on the

Table 1: Comparison of ASW for metrics computed and runtime for scRNA-seq datasets for WSV as compared to tree-based WSV. Euclidean metric for baseline. ∗ indicates method did not converge.

| *Dataset (size)* | PBMC | (2k) | Neurons V1 | (1k) | Lung | (7k) |
|---|---|---|---|---|---|---|
| Metric/method | ASW | Runtime | ASW | Runtime | ASW | Runtime |
| Euclidean | 0.073 | N/A | 0.200 | N/A | 0.046 | N/A |
| Entropic WSV (SSV) | **0.348** | 110 min | 0.256 | 70 min | ∗ | > 12 hours |
| Tree-WSV (ours - 4 iters) | 0.299 | **8 min** | 0.436 | **7 min** | 0.104 | **27 min** |
| Tree-WSV (ours - best) | 0.313 | 72 min | **0.602** | 32 min | **0.457** | 71 min |

incremental furthest search algorithm, resulting in low-depth initial trees. To account for this, we re-ran the Tree-TWD unsupervised ground metric algorithm using the ground metric outputs from the previous run as the distance metric for the initial tree construction. This approach increased the depth of the trees constructed, resulting in improved silhouette scores, but is still entirely unsupervised. Results at 4 iterations and best (max 15 iterations) are summarised in Table 1.

The ASW metrics are comparable in performance to Huizing et al. (2022)'s Sinkhorn singular vectors approach and better than other alternatives for the PBMC dataset, as presented in Table 1 and Appendix G, with Euclidean distance as a baseline. However, the Tree-WSV method vastly outperforms SSV on a similar task for a dataset of labelled neural cells (Tasic et al., 2018) and in the HLCA data (Sikkema et al., 2023). Further, our method has considerably lower runtime. Indeed, on the large HLCA dataset (7000 cells) SSV had not converged after 12 hours. Tree-WSV however can be run on matrices of over 10000 samples/features.

We visualise and validate the cell-type clusters via a 2-component UMAP projection in Figure 3. The clusters for the PBMC dataset are qualitatively similar to those of Huizing et al. (2022) (Fig. 3 A). Importantly, clusters are visually separable from each other.

Intriguingly, in some cases additional sub-clusters are visible further than the initial annotation would suggest: for example, for GABAergic neurons in the Tasic et al. (2018) neuron dataset (Fig. 3 C). This likely indicates a subsequent level of detail in the data: the GABAergic sub-type can have further sub-divisions, notably parvalbumin, somatostatin, VIP and other interneurons (Tasic et al., 2018). These clusters reveal themselves coarsely when we label by them in Figure 3 D. Since Tree-WSV is agnostic to the number of clusters in the data, it is not surprising that more detail can be revealed. This advantageous deeper level of complexity should be investigated further.

Finally, while we did not find that ASW scores had converged after 15 meta-iterations, $w$ vector solutions in the inner power-iteration loop tended to always converge within 10 iterations. Stochasticity inherent from choosing the basis set $U$, as well as the distance used to decide on the initial tree-structure clustering, could both be factors in the meta-algorithm. These are interesting avenues of future theoretical understanding and development that might improve the method.

## 5 Conclusions and perspectives

We present Tree-WSV, a new, computationally efficient approach to learning ground metrics in an unsupervised manner by harnessing the tree-Wasserstein distance as an approximation of the 1-Wasserstein distance. The results on toy and real-world data are fast and scale to larger datasets than previously possible, with favourable results. Furthermore, we provide a new geometric underpinning of the restrictions needed for tree structures to flexibly represent data in low-rank approximations. Our method suggests a fast and recursive algorithm to construct basis sets for pairwise paths in trees which could have wide-spread applicability. While promising, there is a need for further large-scale testing and development of the Tree-WSV algorithm for other use cases, including neuronal activity data. Lastly, understanding theoretical underpinnings of convergence, the role of metrics in designing the initial cluster tree, and stochasticity in the basis set construction could be fruitful avenues for future work.

## ACKNOWLEDGMENTS

K.M.D. and S.H. are supported by the Gatsby Charitable Foundation. K.M.D. was hosted by the Okinawa Institute of Science and Technology (OIST) as a Visiting Research Scholar during this work, and the authors thank OIST for their hospitality. M.Y. is supported by MEXT KAKENHI, grant number 21H04874, and partly supported by MEXT KAKENHI, grant number 24K03004.

The authors thank Geert-Jan Huizing and Aaditya Singh for insightful discussions about the work and early comments. Geert-Jan Huizing also made available the PBMC3k pre-processed dataset from Huizing et al. (2022), which facilitated direct reproduction of and comparison to existing results.

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

## A    PROOF OF PROPOSITION 2.1

From Yamada et al. (2022) and Le et al. (2019), given a tree $\mathcal{T}$ and two probability measures supported on $\mathcal{T}$, the TWD computed using $\mathcal{T}$ is the 1-Wasserstein distance on the tree metric $d_{\mathcal{T}}$. For any two measures, there also exists a tree such that it can approximate the 1-Wasserstein distance between the measures; however, since this is only proved for two measures, combining many measures (samples) as we do in this work is not guaranteed to represent the Wasserstein distance (or indeed any distance) without distortion. In this proof, we seek to show that we can construct a TWD-version of the Wasserstein singular vector problem, but we do not comment on how well this approximates the 1-Wasserstein distance theoretically.

Following Yamada et al. (2022), define a tree parameter-based vector for every $a_i, a_j$ as $\boldsymbol{y}_{i,j}^{(\boldsymbol{A})} :=$
$\boldsymbol{z}_i^{(\boldsymbol{A})} + \boldsymbol{z}_j^{(\boldsymbol{A})} - 2\boldsymbol{z}_i^{(\boldsymbol{A})} \circ \boldsymbol{z}_j^{(\boldsymbol{A})}$ with $\boldsymbol{z}_i^{(\boldsymbol{A})}, \boldsymbol{z}_j^{(\boldsymbol{A})} \in \boldsymbol{Z}^{(\boldsymbol{A})}$, that is $\in \{0,1\}^{N-1}$ (equivalently we will write this as $\mathrm{XOR}\{\boldsymbol{z}_i^{(\boldsymbol{A})}, \boldsymbol{z}_j^{(\boldsymbol{A})}\}$), which is known from the tree structure. The tree metric (distance between any two leaves on $\mathcal{T}_{\boldsymbol{A}}$) is then given by:

$$d_{\mathcal{T}_{\boldsymbol{A}}}(\boldsymbol{a_i}, \boldsymbol{a_j}) = \boldsymbol{w_A}^\top \boldsymbol{y}_{i,j}^{(\boldsymbol{A})}. \tag{6}$$

Note that any vector $\boldsymbol{b_k}$ is just a distribution over leaves ($\boldsymbol{a_i}$) in $\mathcal{T}_{\boldsymbol{A}}$ and vice versa. Using the distribution over the $\boldsymbol{a_i}$ with the ground metric on the matrix/tree $\mathcal{T}_{\boldsymbol{A}}$, we can derive tree Wasserstein distances between any $\boldsymbol{b_k}, \boldsymbol{b_l}$. Since TWD admits the closed form in equation 2 (Yamada et al., 2022), $\mathcal{W}_{\mathcal{T}_{\boldsymbol{A}}}(\boldsymbol{b_k}, \boldsymbol{b_l}) = ||\mathrm{diag}(\boldsymbol{w_{(A)}})\boldsymbol{Z}^{(\boldsymbol{A})}(\boldsymbol{b_k} - \boldsymbol{b_l})||_1$.

Now, let us assume that instead of learning the *ground* metric for the full distance matrix, we want to learn a *tree* metric $d_{\mathcal{T}_{\boldsymbol{A}}}(\boldsymbol{a_i}, \boldsymbol{a_j})$ (6). The equivalent claim as in Huizing et al. (2022) is that we achieve the fixed points in Proposition 2.1. More formally, as before, we would like:

$$\lambda_A \boldsymbol{w_A}^\top \mathbf{Y}^{(\boldsymbol{A})} = \Phi_B(\boldsymbol{w_B}), \qquad \lambda_B \boldsymbol{w_B}^\top \mathbf{Y}^{(\boldsymbol{B})} = \Phi_A(\boldsymbol{w_A})$$

with $\lambda_A, \lambda_B, \tau \in \mathbb{R}_+$, where $\Phi_A(\boldsymbol{w_A})_{i,j} = \mathcal{W}_{\mathcal{T}_{\boldsymbol{A}}}(\boldsymbol{a_i}, \boldsymbol{a_j}) + \tau||\boldsymbol{A}||_\infty R(\boldsymbol{a_i} - \boldsymbol{a_j})$, and $\mathbf{Y}$ is the tensor composed of the $n^2$ (or $m^2$) $\boldsymbol{y}_{i,j}$ vectors. In this case, the $\Phi$s map ground costs to Wasserstein distance matrices.

Let $\boldsymbol{z}_{k,l}^{\boldsymbol{A}} := \boldsymbol{Z}^{(\boldsymbol{A})}(\boldsymbol{b_k} - \boldsymbol{b_l})$. Since $\boldsymbol{w_A} \in \mathbb{R}_+^{N-1}$ is positive, and letting $|\cdot|$ denote element-wise absolute value,

$$\begin{aligned} W_{\mathcal{T}_{\boldsymbol{A}}}(\boldsymbol{b_k}, \boldsymbol{b_l}) &= ||\mathrm{diag}(\boldsymbol{w_A})\boldsymbol{z}_{k,l}^{(\boldsymbol{A})}||_1 \\ &= \sum_{s=1}^{M} (\boldsymbol{w_A})_s |(\boldsymbol{z}_{k,l}^{(\boldsymbol{A})})_s| \\ &= \boldsymbol{w_A}^\top |\boldsymbol{z}_{k,l}^{(\boldsymbol{A})}| \end{aligned}$$

So from (2) and Proposition 2.1:

$$\lambda_A \boldsymbol{w_A}^\top \boldsymbol{y}_{i,j}^{(\boldsymbol{A})} = \boldsymbol{w_B}^\top |\boldsymbol{z}_{i,j}^{(\boldsymbol{B})}|, \qquad \lambda_B \boldsymbol{w_B}^\top \boldsymbol{y}_{k,l}^{(\boldsymbol{B})} = \boldsymbol{w_A}^\top |\boldsymbol{z}_{k,l}^{(\boldsymbol{A})}| \tag{7}$$

$\forall k, l \in \{1, .., m\}, i, j \in \{1, .., n\}$.

In full, the singular vector update becomes to find $\boldsymbol{w_A}$ such that $\forall i, j$

$$\lambda_A \boldsymbol{w_A}^\top \left( \boldsymbol{z}_i^{(\boldsymbol{A})} + \boldsymbol{z}_j^{(\boldsymbol{A})} - 2\boldsymbol{z}_i^{(\boldsymbol{A})} \cdot \boldsymbol{z}_j^{(\boldsymbol{A})} \right) = \left|\left| \mathrm{diag}(\boldsymbol{w_B})\boldsymbol{Z}^{(\boldsymbol{B})}(\boldsymbol{a_i} - \boldsymbol{a_j}) \right|\right|_1 \tag{8}$$

(and symmetrically for the $\boldsymbol{w_B}$ iteration). $\qquad\square$

## B    PROOF OF LEMMA 2.2

The proof proceeds identically to the proof of Theorem 2.3 in Huizing et al. (2022). As in Algorithm 1, the singular value updates satisfy

$$\boldsymbol{w_A}^\top \boldsymbol{U}^{(\boldsymbol{A})} = \frac{\mathcal{W}_{\mathcal{T}_{\boldsymbol{B}}}}{||\mathcal{W}_{\mathcal{T}_{\boldsymbol{B}}}||_\infty},$$

$$\boldsymbol{w_B}^\top \boldsymbol{U}^{(\boldsymbol{B})} = \frac{\mathcal{W}_{\mathcal{T}_{\boldsymbol{A}}}}{||\mathcal{W}_{\mathcal{T}_{\boldsymbol{A}}}||_\infty}.$$

The left-hand sides of these equations are the ground metrics, corresponding to $\mathbb{A}$ and $\mathbb{B}$ in Huizing et al. (2022). The matrices $\boldsymbol{U}^{(A)}$, $\boldsymbol{U}^{(B)}$ are invertible by construction, and so the existence of a fixed point $(\boldsymbol{w_A}^\top \boldsymbol{U}^{(A)}, \boldsymbol{w_B}^\top \boldsymbol{U}^{(B)})$ to the left-hand sides equations above implies the existence of a corresponding fixed point for the weight vectors $(\boldsymbol{w_A}, \boldsymbol{w_B})$.

As the right-hand sides are $L_\infty$-normalised and non-negative, the ground metrics (i.e. left-hand sides) must have all components in the closed interval $[0, 1]$. Hence, reshaping the left-hand sides from $n^2$ (resp. $m^2$)-dimensional vectors to $n \times n$ (resp. $m \times m$) distance matrices, the mapping that updates the ground metrics is a continuous mapping that maps the compact convex set

$$\{(\boldsymbol{C}, \boldsymbol{D}) \in ([0,1]^{n \times n} \times [0,1]^{m \times m}) : C_{i,i} = 0 \ \forall i, \ D_{k,k} = 0 \ \forall k\}$$

to itself. By Brouwer's fixed point theorem, this mapping must have a fixed point. $\qquad\square$

## C   PROOF OF LEMMA 2.3

As preliminaries, we assert that $N > N_{\text{leaf}} = m$, and $\boldsymbol{Z} \in \{0,1\}^{(N-1) \times \ell}$ has rank $m$. Note that $\boldsymbol{Y} \in \{0,1\}^{(N-1) \times \ell \times \ell}$ is strictly a tensor, but in practice we only care for the reshaped matrix $\boldsymbol{Y'} \in \{0,1\}^{(N-1) \times \ell^2}$.

Assume that trees cannot have redundant nodes (i.e. nodes with exactly 1 child, not including themselves), and let every tree have root node $r$ and leaf nodes $1, ..., \ell$, which we also use to index the associated positions along vectors $\boldsymbol{z}$ and $\boldsymbol{y}$.

Note that at most one inner (non-leaf) node can have degree less than 3, and it must be the root. In this case, the root has degree 2. Otherwise all inner nodes are of degree $\geq 3$.

Some bounds for $N$ in terms of $\ell$ can be derived from the observation that rank is upper-bounded by the number of pairs that two distinct leaves can make, noting $\boldsymbol{y}_{i,j} = \boldsymbol{y}_{j,i}$ and $\boldsymbol{y}_{i,i} = 0$. For $\ell \geq 4$,

$$l^2 > \frac{\ell(\ell-1)}{2} \geq 2\ell - 2 \geq N - 1 \geq \ell. \tag{9}$$

Replacing $N - 1$ for $N - 2$ we get bounds for $\ell \geq 2$. $2\ell - 2 \geq N - 1$ in (9) is derived from the upper bound on the number of nodes in the maximal spanning tree, a binary tree.

We prove by induction on the number of tree nodes that the rank of $\boldsymbol{Y'}$ is $N-1$ for any tree structure in which all nodes have degree $\geq 3$, and $N - 2$ otherwise, for any $\ell \geq 2$.

$N - 2$ **base case, $N = 3$:** The tree can only be a root connected to two leaves. There is only one $\boldsymbol{y}$-vector, between the two leaves. So the rank is $N - 2 = 1$.

$N - 1$ **base case, $N = 4$:** $r$ connects to each of the three 3 leaves. In general for $N = \ell + 1, \ell \geq 3$, we can show the rank is $N - 1 = \ell$. Without loss of generality, choose a leaf $i$ and consider $\boldsymbol{y}_{i,j}, \forall j \neq i$. Since every $\boldsymbol{z_j}$ is 0 everywhere except for positions indexed by node $i$ and root $r$, where it is 1, $\boldsymbol{y}_{i,j}$ is 1 at $i, j$ and 0 otherwise. Therefore the set of $\ell - 1$ vectors $\boldsymbol{S_i} = \{\boldsymbol{y}_{i,j}, i \neq j, \forall j\}$ is linearly independent. Additionally, we can choose any two distinct nodes $j, k$ where $j \neq k \neq i$, and $\boldsymbol{S} = \{\boldsymbol{y}_{j,k} \cup \boldsymbol{S_i}\}$ must also be a linearly independent set. This follows because $\boldsymbol{y}_{j,k}$ is 1 at nodes $j, k$ and 0 otherwise; and no linear combination of $\boldsymbol{y}_{i,k}, \boldsymbol{y}_{i,j}$ (with $i = 1$ for both vectors, which should be cancelled) can reproduce $\boldsymbol{y}_{j,k}$.

Can we add any more vectors from $\boldsymbol{Y}$ to $\boldsymbol{S}$ such that the set is still linearly independent? No. To show this, assume that there exists some $\boldsymbol{y}_{g,h} \notin \text{span}(\boldsymbol{S})$. Trivially, $g \neq h \neq i$ and $\{g, h\} \neq \{j, k\}$, so $\boldsymbol{y}_{g,h}$ is 0 everywhere except at the $g, h$ positions. But note that we can write $\boldsymbol{y}_{g,h}$ using vectors in $\boldsymbol{S}$:

$$\boldsymbol{y}_{g,h} = \boldsymbol{y}_{i,g} + \boldsymbol{y}_{i,h} - \big( \underbrace{\boldsymbol{y}_{i,j} + \boldsymbol{y}_{i,k} - \boldsymbol{y}_{j,k}}_{=2 \text{ at } i; \ 0 \text{ otherwise}} \big) \tag{10}$$

So our assumption was incorrect and $\boldsymbol{Y} \in \text{span}(\boldsymbol{S})$. Therefore $\text{rank}(\boldsymbol{Y}) = \ell = N - 1$.

**Induction hypothesis, $N \leq m$:** Assume that the rank of *any* tree structure with parameter $N \leq m, m \in \mathbb{N}$, has rank $N - 1 = m - 1$ if all inner nodes are degree 3 or more, and $N - 2 = m - 2$ otherwise. We call this a $m$-tree.

**Inductive step,** $N = m + 1 \geq 4$**:** We define a set of "adjacent" leaf nodes $\boldsymbol{A}$ to be the set of all leaves connected to the same parent node $\rho$ where $\rho$ has no non-leaf children.

Note first that we can construct any valid $[m + 1]$-tree structure from some previous valid $m$-tree: for any $[m + 1]$-tree, consider a complete set of adjacent leaf nodes of depth at least 2 from $r$ – that is, all the leaves connected to the same non-root node $\rho$ (if depth is less than 2, we get the extended base case). These leaves have common ancestry, and their $\boldsymbol{z}$-parameters differ only at the indices of the leaves themselves. Since $n > 1$, there must exist at least one node $\rho$ that is neither $r$ nor a leaf. Collapse the $\rho$-sub-tree by removing $\rho$ and reconnecting its leaves to the immediate parent node of $\rho$. Then the new tree is an $m$-tree (we have lost one node), as claimed.

Every tree must have at least one such set $\boldsymbol{A}$, and it must have at least 2 elements. We consider the *smallest* $\boldsymbol{A}$-sub-tree and the $\boldsymbol{L}/\boldsymbol{A}$-sub-tree consisting of the rest of the tree, in two cases.

**Case 1,** $|\boldsymbol{A}| \geq 3$**:** From the base case, the rank of the $\boldsymbol{A}$-sub-tree up to the parent node $\rho$ is $|\boldsymbol{A}|$. Now consider the sub-tree consisting of all the other leaves in the tree, $\boldsymbol{L}/\boldsymbol{A}$, up until $\rho$, where the two sub-trees connect. Imagine $\rho$ is a leaf (ignore its children $\boldsymbol{A}$). Then the $\boldsymbol{L}/\boldsymbol{A}$-sub-tree has tree parameter $N_{\boldsymbol{L}/\boldsymbol{A}} = m + 1 - \boldsymbol{A}, \leq m$, so by the inductive hypothesis it has rank $N - 1 - \boldsymbol{A}$ or $N - 2 - \boldsymbol{A}$, depending on the degree of $r$.

How do we join the sub-trees? Let links including $\rho$ now extend to some (the same) new leaf in $\{\boldsymbol{A}\}$, say $a_i$. The union of the $\boldsymbol{y}_{i,j}$ making up each sub-tree's basis set is linearly independent. To see this, note that the only $\boldsymbol{y}_{i,j}$ that have a 1 at position $\rho$ were those ending at $\rho$ the leaf on $\{\boldsymbol{L}/\boldsymbol{A}\}$. So to express any $\boldsymbol{y}_{l_k,a_i}$, a link via $\rho$ is needed, that is the rank is lower-bounded by $N - 1$ or $N - 2$, respectively.

We cannot add any more $\boldsymbol{y}$s: since the sub-trees are saturated by assumption, any addition $\boldsymbol{y}$ should also be a link, but any link can be expressed via an existing link through $\rho$ and combinations of the sub-tree basis sets (following the argument for the extended base case $N > 3$, since we know that each sub-tree has more than 3 nodes). Therefore rank($[m + 1]$-tree) $= N - 1$ or $N - 2$ based on the sub-tree, as required.

**Case 2,** $|\boldsymbol{A}| = 2$**:** In this case, the $\boldsymbol{L}/\boldsymbol{A}$-sub-tree has rank $N - 3$ or $N - 4$ and the $\boldsymbol{A}$-sub-tree has internal rank 1 since there is just one pair of leaves to make $\boldsymbol{y}_{a_1,a_2}$. Without loss of generality, assume the $\rho$ links extend to $\boldsymbol{a_1}$, with $\{\boldsymbol{y}_{\ell_i,a_1}\}$ for some (not necessarily single) $i$, necessarily linearly independent of $\boldsymbol{y}_{a_1,a_2}$. The total rank is then $N - 2$ or $N - 3$.

In the 2-case, we can also create another $\boldsymbol{y}_{l_j,a_2}$, where $j$ can either be the same as $i$ or different.

How do we know that $\boldsymbol{y}_{l_j,a_2}$ is not in the span of the two sub-tree's $\boldsymbol{y}$ vectors? Assume it is in the span. $\boldsymbol{y}_{l_j,a_2}$ has 1s at $\rho$. The only other $\boldsymbol{y}$s which shares this property are from $\boldsymbol{y}_{l_i,a_1}$. So, if $\boldsymbol{y}_{l_j,a_2}$ could be written using already chosen $\boldsymbol{y}$s, its decomposition must include $\boldsymbol{y}_{l_i,a_1}$. But since $\boldsymbol{a_1}$ is known to be zero in $\boldsymbol{y}_{a_2,l_j}$, but is non-zero in the path containing $\boldsymbol{y}_{l_i,a_1}$, $[\boldsymbol{a_1}] = 1$ (where $[\boldsymbol{a_j}] = 1$ refers to the component of $\boldsymbol{y}_{l_i,a_1}$ corresponding to $\boldsymbol{a_j}$) should be "removed" through linear combinations with other $\boldsymbol{y}$s; however, there is just one $\boldsymbol{y}$ involving $\boldsymbol{a_1}$, which is $\boldsymbol{y}_{a_1,a_2}$. So we must subtract $\boldsymbol{y}_{a_1,a_2}$, resulting in a value of $-1$ at $[\boldsymbol{a_2}]$. But we require $[\boldsymbol{a_2}]$ to be 1: there is no linear combination that allows both $[\rho]$ and $[\boldsymbol{a_2}]$ to be 1.

So we must be able to add one link, and so here, too, the rank is $N - 1$ or $N - 2$ in total (trivially any further $\boldsymbol{y}$s must be links and so would be in the span).

$\square$

# D  RECURSIVE ALGORITHM FOR TREE BASIS CALCULATION

---
**Algorithm 2** Tree basis set recursion

---
 **basis_tree**
 *INPUTS* : *path_matrix, $U$ basis – initially a tree parameter/parent matrix, rank – initially 0*
 $N, L \leftarrow$ shape of *path_matrix*{nodes, leaves} ($U$ has shape $N \times N$)
 **if** $N == 3$ (base case: 2 leaves and 1 root) **then**
  $U[\text{rank}] \leftarrow$ path of the 2 leaves (perform XOR on 2 path_matrix columns), rank $\leftarrow 1$
 **else**
  smallest-parent $\leftarrow \arg\min_{\text{row index}} \sum_{\text{rows}} U[L:]$, size $\leftarrow \sum U[\text{smallest-parent}]$
  smallest-parent's-leaves $\leftarrow$ non-zero indices of smallest-parent row
  **if** size $== 2$ (2 adjacent leaves) **then**
   $U[\text{rank}] \leftarrow$ path of smallest-parent's-leaves (perform XOR on 2 path_matrix columns)
   $U[\text{rank+1}] \leftarrow$ path of 1 smallest-parent's-leaf & a leaf $\notin$ {smallest-parent's-leaves}
   rank$+ = 2$
  **else**
   $U[\text{rank}] \leftarrow$ path of smallest-parent's-leaves 0,size $- 1$
   rank$+ = 1$
   **for** $i = 0$ to size $- 2$ **do**
    $U[\text{rank} + i] \leftarrow$ path of smallest-parent's-leaves$i, i + 1$
   **end for**
   rank$+ = n - 1$
  **end if**
  **for** $i$ in smallest-parent's-leaves$[1:]$ **do**
   path_matrix column at smallest-parent's-leaves index $i \leftarrow 0$ (i.e. discount all leaves but one for the next recursion – can be made stochastic)
  **end for**
  $U, \text{rank} \leftarrow \text{basis\_tree}(\text{path\_matrix}, U, \text{rank})$
 **end if**
 **return** $U, \text{rank}$

---

# E  EMPIRICAL CONVERGENCE

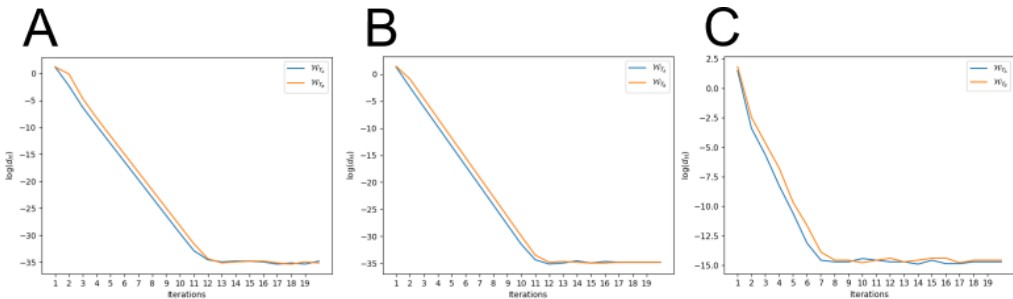

Figure 4: Empirical convergence (Hilbert metric) observed for different toy torus dataset sizes across power iterations. The smaller dimension is shown in blue: (**A**) 60 x 80 (with SVD basis set calculation); (**B**) 100 x 200 (with SVD basis set calculation); (**C**) 1500 x 300 (with Algorithm D basis set calculation).

# F  DETAILS ON GENOMICS DATA PREPROCESSING AND EXPERIMENTS

## F.1  DATA PREPROCESSING

All single-cell RNA sequencing data was processed using standard protocols, including CPM-normalisation, log1p-transformation, and selection of genes based on highest variability. Canonical markers were added according to source.

PBMC3K data was recovered using Scanpy (Wolf et al., 2018). Preprocessing procedure/code was provided on request from Huizing et al. (2022) and included cell type annotation using Azimuth marker genes (Hao et al., 2021), sampling only annotations for which confidence was over 90%. The results of SSV on this dataset were identical to that reported previously (Huizing et al., 2022).

Neural cell data and labels from Tasic et al. (2018) were recovered from the NCBI GEO repository, Series GSE71585 (Clough et al., 2024) and annotated using the Allen Brain Map (Lein et al., 2007) and CZ CELLxGENE (Abdulla et al., 2023).

The Human Lung Cell Atlas (HLCA) was recovered, and annotated, from CZ CELLxGENE (Abdulla et al., 2023), and originally published by Sikkema et al. (2023). Because taking a subset might disrupt the original annotation, low-gene-count cells and cells with labelling entropy score less than 0.1 were removed.

### F.2    ALGORITHM PARAMETERS

Genomics experiments using Tree-WSV were run for 20 iterations of the internal linear system of equations singular vector loop (finding $w$) and 15 meta-iterations of the entire algorithm (starting with constructing new trees based on the 2 previous weight matrices, through to computing 2 new TWD-based distance matrices). A JAX random seed of 0 was used. The ASW and total runtime after the 4th meta-iteration and the best score overall (usually the 12-15th meta-iteration) were reported.

Genomics experiments using SSV were run for 15 iterations of the singular vectors loop with $\tau = 0.001, \epsilon = 0.1$; these follow the settings and replicate the results in Huizing et al. (2022).

### F.3    COMPUTATIONAL RESOURCES

Computations on CPU were done using a Apple M2 MacBook Air M2 with 16 GB RAM. GPU computations were performed on on an NVIDIA A100 node with 16 GB requested memory.

## G    COMPARISON TO OTHER METHODS FOR IMPROVING CELL-TYPE CLUSTERING

We reproduce the table from Huizing et al. (2022) showing other methods that can be used to cluster cell-type data from scRNA-seq experiments as compared to ours on the PBMC dataset.

Table 2: Comparison of average silhouette width (ASW) for distance metrics computed and runtime in the PBMC dataset.

| Dataset (size) | PBMC | (2k) |
|---|---|---|
| Metric/method | ASW | Runtime |
| PCA / $\ell^2$ | 0.238 | |
| Kernel PCA / $\ell^2$ | 0.241 | |
| scVI embedding / $\ell^2$ | 0.168 | |
| Sinkhorn | 0.003 | |
| Gene Mover Distance | 0.066 | |
| Euclidean | 0.073 | |
| Entropic WSV (SSV) | **0.348** | 110 min |
| Tree-WSV (ours - 4 iters) | 0.299 | **8 min** |
| Tree-WSV (ours - best) | 0.313 | **72 min** |

