# OpenReview forum: "Fast unsupervised ground metric learning with tree-Wasserstein distance"
_ICLR.cc/2025/Conference — ICLR 2025 Poster_

### Official Review · Reviewer_SCu1 · 2024-10-29

**Soundness:** 3
**Presentation:** 3
**Contribution:** 3
**Rating:** 8
**Confidence:** 3

**Summary:**

The paper proposes an interesting variation of Wasserstein singular vectors by embedding samples and features on a tree. By doing so, they claim to have achieve a cubic complexity as opposed to quintic complexity of the standard method. While the authors show interesting results, I believe the manuscript can be accepted after a major revision.

**Strengths:**

I think the paper is proposing a novel idea for estimating ground metric learning of unlabeled data set, that reduces the complexity of each iteration from $\mathcal{O}(N^5)$ to $\mathcal{O}(N^3)$.

**Weaknesses:**

Some important details of the proposed method are missing in the paper. This includes convergence rate, well-posedness, and memory consumption of the proposed method. See my questions/comments below.

**Questions:**

**Major**:

- How is the condition number of eq. 5? Does the system become ill-conditioned as matrix becomes large?
- Section 2.3, what is the convergence rate of the proposed iterative method? How it is affected by the data set size?
- What is the complexity of Cluster Tree used here?
- Please add details of memory consumption for your proposed iterative algorithm and approximation to SVD.
- I think the authors should make more effort to show the claimed complexity $\mathcal{O}(N^3)$. Consider one of the data set, and test the method against benchmark for a range of $N$.

**Minor**:

- In abstract, "...a fast and recursive algorithm..." and not "a fast, recursive algorithm…”
- In abstract, what does "low-compute application" mean?
- P2, l82: "... to have a solution" and not "to have solution"
- P3, l108: Is there a guarantee that the shortest path between any two nodes on a tree is unique?
- Figure 1: Caption refers to b5, but I don’t see b5 in the graph.
- In the references, make sure to use capital letters when needed, e.g. use “Wasserstein” instead of “wasserstein”

---

> ### Author Response · Authors · 2024-11-21
> **Reviewer 4: Improved submission addressing convergence, well-posedness and memory, and details on condition numbers**
>
> We were encouraged to read that the reviewer believes the paper can be accepted after a major revision, and we hope the revised manuscript will meet these criteria, particularly addressing the topics of convergence rate, well-posedness and memory consumption. We thank the reviewer for their clear revision asks.
>
> Major:
>
> * The condition number on eq. 5 (now rewritten as new eq. 5, the complete linear system of equations, to which I assume the reviewer is referring) is good (near 1) for the SVD approach, and we believe acceptable for the recursive approach (but we would appreciate any further insights or reinterpretation, as this is not our field of expertise!). For the SVD approach, the condition numbers on $\boldsymbol{U}$ are all very close to 1 with error on the order of 1e-15 (these were computed for (torus) dataset sizes 80 x 60, 100 x 200 and 500 x 500 using numpy.linalg's implementation of the condition number and the standard 2-norm). For the recursive algorithm, although we get full-rank matrices, the condition number is higher, on the order of 1000-3000 for sample sizes of 1000 and 2000 respectively. While this is much larger than 1, it should not affect accuracy within a margin of 1e-5 for the $\boldsymbol{w}$ vectors (based on a 1e-8 numerical precision of jax.float32) -- and these are generally around 1e-3 for large n. As the method scales larger, this could of course be a problem, and one option could be to compute multiple recursive basis sets and either take one with lowest condition number or iterate Tree-WSV with several and average. Thanks for the suggestion to confirm these! Would the reviewer recommend these numbers be included in the paper as a comment on the least squares solver's convergence?
>
> * Regarding convergence rate, we now include empirical convergence (of power iterations) for various dataset sizes 80 x 60, 100 x 200 and 1500 x 300 in Appendix E. In practice, we always observed fast (within 10 iterations) convergence of power iterations, and built-in scipy and jax.numpy least squares / NNLS solvers are presumed to have converged with high accuracy based on the condition numbers. Convergence appears stable with larger dataset sizes.
>
> * ClusterTree implemented in this way has complexity $\mathcal{O}(n\kappa)$ where $\kappa$ is the number of leaf-clusters (in our case set automatically by the hyperparameter controlling the depth of the tree) [1] (Gonzalez, 1985). We have added this to the paper.
>
> * Regarding memory consumption of sparse SVD and the iterative algorithm, we have added the comment: "Because this operation requires computing a large tensor $\boldsymbol{\mathsf{Y}}$ reshaped into a long rectangular matrix of size $(N-1) \times n^2$, the sparse method does not scale well with large $n,m$ from a memory-consumption point of view." We elaborate on this in our response to reviewer 3. In summary, (sparse) SVD has complexity cubic in $n$ (resp. $m$), whereas our iterative algorithm has worst-case complexity quadratic in $n$ (resp. $m$). We observe that for small $n,m$, sparse SVD is faster, whereas for large $n,m$, our iterative algorithm is faster. We have not explained in this detail in-text besides the comment on memory consumption, but can do so if you feel it would help.
>
> * In terms of showing complexity, we have been more cautious in explaining the calculation in section 2.3.2 ("This gives overall complexity per power iteration: $\mathcal{O}\left(N^3+mN+M^3+nM\right) < \mathcal{O}\left(n^3 + m^3 + mn\right)$, using $N<2n-1, M<2m-1$.") We could certainly also compare runtime for a variety of different dataset sizes if the reviewer feels it would help our paper a lot, but we hoped that including the runtime after a number of meta-iterations for each dataset in Table 2 (each of which ran for a similar number of inner power iterations) would provide some estimate for the interested reader. Exact time complexity curve fits will be limited to the exact number of edges ($N$) in any tree construction, too. Does the reviewer feel that including a time complexity fit in the appendix on the number of edges would improve the paper greatly? If so, we can produce one.
>
> [1] Teofilo F. Gonzalez. Clustering to minimize the maximum intercluster distance. Theoretical Computer Science, 38:293–306, 1 1985. ISSN 0304-3975. doi: 10.1016/0304-3975(85)90224-5

---

> > ### Author Response · Authors · 2024-11-21
> > **(Part 2) Minor points**
> >
> > (Part 2)
> >
> > Addressing minor points:
> >
> > * The abstract now reads "a fast and recursive..." (there were a few other places that also needed correction in this way).
> > * Low-compute application means application without requiring high GPU etc time. However, since this could imply something too specific, we now leave out the word "low-compute."
> > * P2, l82: "... to have a solution" and not "to have solution" -- corrected, thanks!
> > * P3, l108: Is there a guarantee that the shortest path between any two nodes on a tree is unique? -- Yes, there is. In fact there is only one path between any two leaves on a tree without loops.
> > * Figure 1: We added a $b_5$ node, thanks!
> > * The references have now all been rigorously checked and "Wasserstein" capitalised, apologies for this error!
> >
> > We thank the reviewer again for their read, and extra expertise regarding condition number -- we would appreciate any further comments on this aspect. We hope the extra explanations regarding memory and convergence are helpful!

---

> > ### Comment · Reviewer_SCu1 · 2024-11-27
> >
> > I thank the authors for their answers and rebuttal. I believe the manuscript has improved a lot. I increased my initial score.
> >
> > > Would the reviewer recommend these numbers be included in the paper as a comment on the least squares solver's convergence?
> >
> > I think adding some details about the condition number improves the paper and helps the reader.
> >
> > > We observe that for small $n,m$, sparse SVD is faster, whereas for large $n,m$, our iterative algorithm is faster. We have not explained in this detail in-text besides the comment on memory consumption, but can do so if you feel it would help.
> >
> > I see analogy with iterative versus direct solvers of linear system of equations. I guess this is a common knowledge and therefore, I agree with authors to leave it out.
> >
> > > Does the reviewer feel that including a time complexity fit in the appendix on the number of edges would improve the paper greatly? If so, we can produce one.
> >
> > I think having the time complexity in the appendix would certainly help the reader as a point of reference. However, I leave it to the authors to make that judgement.

---

> > > ### Author Response · Authors · 2024-11-29
> > > **Thanks; and we will add the condition number and consider time complexity**
> > >
> > > We thank the reviewer for their comment and improved rating.
> > >
> > > As suggested, we shall add details about the condition number in-text. We shall consider adding in time complexity fits in the appendix (we would like to run a few versions of this with different $n,m$ first to see if it will be helpful).

---

### Official Review · Reviewer_pG9e · 2024-11-01

**Soundness:** 3
**Presentation:** 3
**Contribution:** 2
**Rating:** 6
**Confidence:** 3

**Summary:**

The paper presents a novel approach for unsupervised ground metric learning using Tree-Wasserstein Distance (TWD) as a low-rank approximation of the computationally intensive Wasserstein Singular Vector (WSV) method. The proposed method embeds samples and features in tree structures, reducing the computational complexity from O(n⁵) in traditional WSV to O(n³) by learning distances between data points as TWD on trees. Empirical results indicate that the method achieves similar or better clustering accuracy compared to Sinkhorn singular vectors (SSV) while maintaining much faster runtimes.

This paper is the improved version of the workshop paper “Unsupervised Ground Metric Learning with Tree Wasserstein Distance”. The primary innovation of this work is adding recursive basis set computation for tree-based WSV.

**Strengths:**

1. The paper offers rigorous theoretical support for the TWD method, with proofs on the uniqueness and existence of solutions within specific tree configurations.
2. The empirical results are presented clearly, with comparative metrics that directly illustrate the computational runtime saving and clustering performance.
3. The paper provides a solid background review on optimal transport theory and the tree-Wasserstein distance.

**Weaknesses:**

1. Although the ClusterTree algorithm plays a significant role in the tree structure initialization, there is limited background provided on how it operates, what assumptions it makes, or its typical applications. I reviewed both references—Le et al. (2019) and Indyk & Thaper (2003)—but did not find any mention of a ClusterTree. Could the author be referring to the ‘Partition_Tree_Metric’ described in Le et al. (2019)?
2. The algorithm section mentions differences in handling ‘large’ and ‘small’ datasets but does not specify the boundary between the two. What happens if either m or n is very large while the other meets the ‘small’ criteria?
3. The paper’s notation is sometimes inconsistent, making it challenging to reference equations or terms precisely. For example, on line 201, a_i and a_j are bold, but on line 205, a is not bold. On line 211, what does the cost matrix B represent?
4. Figure 3 could be better organized. The paper does not provide a comparison of how other metrics perform on these datasets.
5. Line 557, The URL is invalid.

**Questions:**

Refer to weaknesses.

---

> ### Author Response · Authors · 2024-11-21
> **Reviewer 3: Improved submission addressing ClusterTree, dataset size, and notation**
>
> (Part 1)
>
> We thank the reviewer for the response and for recognising the paper's strengths (including providing rigorous theoretical support for our TWD method, clear empirical results and a review of the related literature). We also agree that the paper is indeed an improved version on the workshop paper mentioned; however, since the workshop was non-archival, we would ask the reviewer to evaluate entirely and independently for this submission to ICLR (and, indeed, our read of the review supports that the reviewer has done this!).
>
> In terms of the weaknesses:
>
> * We apologise for not providing adequate background regarding ClusterTree. The details are indeed equivalent to the method in Le et al. (2019), although they do not yet refer to the method as ClusterTree; this was adopted by later authors who reference their work. It is not quite the same as Algorithm 1 in that paper but rather the extension described in the next paragraph (using Alg. 2). Further background on clustering to form trees (and the Farthest-Point Clustering Algorithm) are provided in their main-text and appendices. We have added a few sentences about this and referenced earlier authors, but would like to stress in reply that the exact clustering tree algorithm does not make much difference in our hands, since we learn edge weights. We have compared with using QuadTree for initialisation, although we prefer ClusterTree as there is more flexibility in the number of leaves per cluster. Note that Indyk and Thaper extend QuadTree to higher dimensions, but we agree the reference in that point in text was unclear -- we now also reference the original QuadTree paper and Indyk and Thaper as a follow-on. Here is how we have re-introduced ClusterTree: "Note that here, ClusterTree is a modification of QuadTree (Samet (1984) and extended to higher dimensions via a grid construction in Indyk & Thaper (2003)). Our method follows the implementation in Le et al. (2019) and Gonzalez (1985). ClusterTree implemented in this way has complexity $\mathcal{O}(n\kappa)$ where $\kappa$ is the number of leaf-clusters (in our case set automatically by the hyperparameter controlling the depth of the tree) (Gonzalez, 1985)." Also see [1]. We based our implementation of ClusterTree on the following publicly available repository and would be happy to include it in the paper if you feel it would strengthen the background on tree construction: https://github.com/oist/treeOT
>
> * We agree that Algorithm 1 was scanty on ``bounds'' for large versus small datasets and we have now included some. This is really to do with memory issues when using SVD on large datasets. Because the sparse SVD method requires as input the entire $\boldsymbol{Y^\prime} \in \mathbb{R}^{(N-1) \times n^2}$ matrix, when $n$ is very large that matrix becomes prohibitive to store; in addition SVD itself scales poorly in terms of memory allocation (see https://fa.bianp.net/blog/2012/singular-value-decomposition-in-scipy/). Standard SVD with an input matrix of size $p \times q$ with $p>q$ has complexity $\mathcal{O}(pq^2)$, and sparse SVD should be similar, although we cannot find exact complexity references. Hence, as $n < N < 2n-1$, SVD on the matrix $\boldsymbol{Y^\prime}$ has complexity cubic in $n$. By our estimation, the recursive algorithm is fast: it recurses until at most depth $n$ (the number of leaves), and each recursion can include at most $n$ operations, so the total worst-case complexity is quadratic in $n$ (and symmetrically for $m$). In terms of the latter question of whether one or the other of $n,m$ is larger: separate methods could be used to compute each of the two basis sets, since there is no interaction between the trees. Towards the main points of our work, either method is fine and perhaps it would be simpler to present all work using the recursive algorithm. However, since it performs slower for smaller $n,m$ empirically, and since we felt that this was easily addressed with a more standard approach than the newly introduced algorithm, we wanted to include both. If one of $n,m$ is small and the other is large, we therefore suggest to use SVD on the small one and our recursive algorithm on the large one. There also may be some reasons to prefer SVD for its condition number (see comment by reviewer 4). We are interested to know what you think given this extra background -- please let us know!
>
> * We have made the notation consistent throughout, using ICLR guidelines to bold italicise vectors and matrices respectively. We believe this is now all consistent and apologise for the poor notation in the submission rush before! We have also restated Theorem 2.2 as a lemma and a theorem which we believe will help to make sense of notation, and expanded on notation definitions throughout. Notation in section 1.1. is consistent with 2.1.
>
> [1] Teofilo F. Gonzalez. Clustering to minimize the maximum intercluster distance. Theoretical Com-
> puter Science, 38:293–306, 1 1985.

---

> ### Author Response · Authors · 2024-11-21
> **Cont: references, Figure 3 and other metrics**
>
> (Part 2)
>
> * The cost matrix on l. 211 should really read $C$, that is a typo (we have updated that section to aid explanation more generally). This matches the general definition for Wasserstein distance in section 1.1, and is illustrative in this paragraph.
>
> * Figure 3 has been reorganised to be neater, which we hope offers an improvement.
>
> * Regarding other metrics, do you mean other clustering metrics like Adjusted Rand Index, Davies-Bouldin? Or other methods (metric learning approaches) like those in the appendix table, but extended to the other datasets? If you have suggestions on particular preferred comparison metrics, we would be happy to include with this clarifying point.
>
> * All references have been carefully checked and the URL previously on l.557 (Le et al., 2019) now points to the official, valid URL.
>
> Overall, we thank the reviewer in particular for noting the need to review and explain ClusterTree and the SVD versus recursive algorithm approaches, since these are indeed core to the Tree-WSV approach. We look forward to discussing any ongoing concerns and suggestions further.

---

> > ### Comment · Reviewer_pG9e · 2024-11-30
> >
> > Thank you to the authors for addressing my concerns. I am satisfied with the clarifications provided and have updated my score accordingly.

---

### Official Review · Reviewer_HU6t · 2024-11-02

**Soundness:** 3
**Presentation:** 3
**Contribution:** 3
**Rating:** 8
**Confidence:** 3

**Summary:**

The authors apply the unsupervised metric learning algorithm from Huizing et al. (2022) (which used Wasserstein and entropy regularized Wasserstein) to learn ground metrics for spaces of histograms based on the Tree Wasserstein distance.

**Strengths:**

The idea is sound, and there is a significant element of originality, especially in the development of the algorithm in Appendix C.

**Weaknesses:**

The paper seems rushed overall, there are a large number of typos and a number of results that should have been presented as Theorems are merely stated informally.

1. l.238 – 241 these statements require a proof. Especially the convergence, since it was already somewhat delicate in Huizing et al. (2022).
2. l.263 “Wasserstein” -> “Tree Wasserstein” or else requires a proof.
3. I find Theorem 2.2 hard to interpret. Can the authors rephrase the interpretation in the next paragraph (l. 256-l.260) as a Theorem and include the current Theorem 2.2 as a Lemma?
4. l.354 The previous work used n=100, m=80 and not n=80, m=60 as this paper states.
5. l.367 Why are you using a different metric (Frobenius norm) than the original work? How does your method compare when using the same metric d_h(B, B’) = ||log(B/B’)||_V?

**Questions:**

List of typos. I suggest that the authors give the manuscript a thorough proof-reading.

* l.39 confusing, why is the optimal sample distance the Wasserstein distance?
* l.106 in what sense is SWD a “geometric embedding”?
* l.111 definition of pi?
* l.116 what does “good approximation” mean here?
* l.140 precise the meaning of normalized.
* l.142 distribution -> “probability distribution”
* l.150 R and tau are undefined
* l.150 not clear what \Phi_A is. Huizing et al. (2022) describe Phi_A as ”lifts a ground metric to a pairwise distance matrix”. The authors need to explain the definition of Phi_A.
* l.151 equivalence to (3) assumes tau = 0. The authors need to be a bit more careful in their recap of Huizing et al. (2022).
* l.157 not sure that the remark in parentheses is correct, according to Huizing et al. (2022) a single Wasserstein iteration is n^3 log n. Also how many distances do we compute when we compute “m^2, n^2 W. distances”? Is is m^2 + n^2, m^2 * n^2, something else?
* l.201 W_{T_B} instead of W_B ?
* l.205 z_i^(A) undefined
* l.205 Z^{(B)} or Z_B?
* l.214 W_{TB} -> W_{T_B}

---

> ### Author Response · Authors · 2024-11-21
> **Reviewer 2: Improved submission addressing theoretical rigour, typos, and specific points**
>
> (Part 1: Weaknesses)
>
> We thank the reviewer for recognising the originality and for their interest in the algorithm in the Appendix, about which we were particularly happy. We apologise for the rushed appearance and assure the reviewer that we have carefully proof-read, corrected typos, and rewritten the manuscript accordingly.
>
> Please note the general updates across reviewers, which include:
> * A careful re-read and incorporation of the theory in Huizing et al., including a new Lemma about existence of solutions for power iterations, and more caution regarding convergence guarantees.
> * A full attempt to standardise notation and remove typos.
> * Rephrasing Theorem 2.2 (thanks so much for this suggestion!)
>
> In particular regarding weaknesses:
> * Prev. l. 238-241: We now address more rigourously one of the guarantees from Huizing et al. in Lemma 2.2 (showing that the SV problem / power iterations has a solution). We believe that we could extend the argument from Huizing et al. using a large $\tau$ to support uniqueness and convergence of the SV solution in our case, too. For now, we show empirical convergence with the same linear dynamics as in Huizing et al. for different dataset sizes (implicitly for $\tau=0$) in Appendix E. In practice, we always observed quick convergence of power iterations.
> * In prev. l. 263 "Wasserstein" is now replaced with `"tree-Wasserstein" (we agree with the reviewer on this point completely: in particular, the paper by Yamada et al. (2022) [1] showing TWD approximates the 1-WD does so mostly via empirical results, and while the equivalent empiric results are strong, we do not know a convincing proof of the bound of the approximation).
> * We have rephrased Theorem 2.2 as a Lemma and the next paragraph as a Theorem (with some more rigour in the proof), as suggested -- we think that this was an excellent suggestion that helps to both formalise and clarify the text. Thank you! The lemma is identical to previous Theorem 2.2. The Theorem reads (sic): "Given any tree $T_A$ with leaves the rows of a data matrix $X$ such that the root node of $T_A$ has degree 3 or more, and a tree $T_B$ with leaves given by $B$ the columns of $X$, there exists a unique, non-zero solution for $\boldsymbol{w_A}$ in the system of linear equations."
>
> * We agree that the sizes of the experimental datasets are slightly different. Note that we get similar results on the same dataset size ($n=100,m=80$) and would be happy to replace the figure with one using these dimensions if the reviewer feels it would strengthen the submission (we used the smaller just because it is faster to run the full WSV method multiple times). However, we also included row/column permutations of the matrix, so the dataset is already considerably different from the original. To clarify that this is the case, we replaced "as employed in previous work" to "modified from prior work" in the first paragraph of section 3.
>
> * Regarding the different metric: we compare the final distance matrices between methods, as compared to the distance matrices found with WSV. We thought that the Frobenius norm is appropriate since it penalises each matrix element, whereas the Hilbert metric considers upper and lower bound logarithmic differences between the two matrices (nonetheless useful as these bounds get closer, i.e. for convergence). Is there another reason to prefer the Hilbert metric? Note that our results are broadly similar when using the Hilbert metric $d_H$: for example, on the same $60 \times 80$ dataset, Tree-WSV scores using $d_H$ are 1.38-1.41 while SSV scores are 21.2-21.7. If the reviewer would like us to replace the accuracy scores in Fig. 2 with this metric, we are happy to do so.
>
> [1] M. Yamada, Y. Takezawa, R. Sato, H. Bao, Z. Kozareva, and S. Ravi. Approximating 1-Wasserstein
> distance with trees. Transactions on Machine Learning Research, 2022. ISSN 2835-8856. URL
> https://openreview.net/forum?id=Ig82l87ZVU.

---

> > ### Comment · Reviewer_HU6t · 2024-11-27
> >
> > Thank you for the revised manuscript, it looks much better now.
> >
> > I think it would be useful to provide the data for the Hilbert metric used in Huizing et al. to enable a direct comparison between the methods. For instance, imagine somebody wants to prove convergence bounds in terms of $d_H$, they might want to know what is the best they can hope to achieve for TWSV compared to WSV.
> >
> > Another minor comment on the form: is there a reason why you don't spell out "without loss of generality"?
> >
> > I raise my rating to accept.

---

> ### Author Response · Authors · 2024-11-21
> **(Part 2) Addressing questions**
>
> Questions:
>
> * We have addressed the typos. Thank you for your careful read!
>
> * l. 39 previously called Wasserstein distance an optimal sample distance. Of course, this is a heuristic used by us to introduce OT, and the reality is more subtle (a metric that takes into account both mappings (weightings of features) and distances between features). We have tried to rephrase this as: "$k$-means gives equal weighting to the distance between each pair of features. From an optimal transport theory perspective, it could be argued that a more holistic sample distance is given by the Wasserstein distance: ..."
>
> * l. 106: SWD is a geometric embedding insofar as it is a projection. We have rephrased this part and incorporated it in the general computational complexity background section: "Sliced-Wasserstein distance (SWD), in which $\mathcal{W}_{C}$ is computed via projection to a one-dimensional subspace, improves on this complexity to $\mathcal{O}(n\log{n})$ (Kolouri et al., 2016). SWD can be expressed as a special case of a geometric embedding, tree-Wasserstein distance (TWD), when the tree structure comprises a root connected to leaves directly (Indyk & Thaper, 2003; Le et al., 2019)."
>
> * l. 111: pi is the transport plan. We now define it (as well as U, the set of joint probability distributions): "Given a tree metric $d_\mathcal{T}$ on leaves $x,y$ and transport plan $\pi$ between measures $\mu,\nu$, the TWD can be written as (previous expression, does not render here), where $U (\mu, \nu ) =$ (full expression inserted in-text, does not render here) is the set of joint probability distributions with marginals $\mu$ on $\mathcal{X}$ and $\nu$ on $\mathcal{Y}$ ."
>
> * l. 116. Goodness of approximation of TWD to 1-WD: This is a good question. There are theoretical results that guarantee there exist trees that can fully express the 1-WD, and empirical results that show that learning edge weights provides good approximation -- see Yamada et al. 2022. We have updated to include ``TWD is a good empirical approximation ...'' An avenue for future work could include seeing whether our theory (specifically Lemma 2.3 / Theorem 2.4) could help to provide bounds on the approximation, via the constructive proof in Yamada et al. (2022).
>
> * l. 140: normalized = sums to 1 (added).
>
> * l. 142: changed.
>
> * l. 150: $R, \tau$ are now defined, and $\Phi_{\boldsymbol{A}}$ described.
>
> * l. 151: equivalence now states that $\tau = 0$, thank you.
>
> * l. 157: We believe the remark in parentheses is correct, but have clarified it to read "(since it requires computing $m^2$ Wasserstein distances where each distance is $\mathcal{O}(n^3\log(n))$ then $n^2$ Wasserstein distances where each distance is $\mathcal{O}(m^3\log(m))$)". To explain: Huizing et al. (2022) state that a single power iteration is $\mathcal{O}(n^2m^2(n \log(n) + m \log(m)))$ for $n$ samples and $m$ features. Our comment in parentheses was to explain that calculation: each Wasserstein distance on $\mathbb{R}^n_+$ is computed in $n^3\log(n)$ and there are $m^2$ of these to compute in the distance matrix, whereas on $\mathbb{R}^m_+$ there are $n^2$ distances to compute and $m^3\log(m)$ for each. Summing and factorising you get the power iteration complexity. Note that we have changed the abstract to use this exact quantity rather than a (loose) approximation.
>
> * l. 201: We agree and have replaced $\mathcal{W}_{T_B}$ instead of $\mathcal{W}_B$.
>
> * l. 205: We now define $\boldsymbol{z_i^{(A)}}$ in the theorem: "where $\boldsymbol{z_i^{(A)}}$ is the $i$th column of $\boldsymbol{Z^{(A)}}$."
>
> * l.205: Should read $\boldsymbol{Z^{(B)}}$, thanks!
>
> * l.214: Typo fixed, thanks!
>
> To reiterate, the idea to restructure the previous theorem as a lemma and theorem and to carefully review Huizing et al. helped to formalise the theory and improve the manuscript greatly -- thank you for this! We look forward to any further engagement or ideas you might have.

---

> ### Author Response · Authors · 2024-11-29
> **Thanks; and we will add the Hilbert metric**
>
> We thank the reviewer for the comment and improved rating in response.
>
> The reasoning to include the Hilbert metric to aid comparison / convergence proofs makes sense -- we will add that in for Figure 2 (we already adopted it for the convergence figure in the appendix). We are also happy to spell out without loss of generality (indeed, we ought not assume this a standard abbreviation!).

---

### Official Review · Reviewer_1BGM · 2024-11-03

**Soundness:** 3
**Presentation:** 1
**Contribution:** 2
**Rating:** 6
**Confidence:** 4

**Summary:**

The paper introduces Tree-WSV, integrating TWD with WSV (Huizing et al., 2022).

**Strengths:**

The proposed method demonstrates efficiency compared to WSV and SSV presented in Huizing et al. (2022).

**Weaknesses:**

- The authors missed important related works [1, 2, 3, 4, 5] that consider the relationships between the samples are informed by the relationships between the columns, and vice versa, for Wasserstein distance and specifically in tree-related settings [2,5]. Specifically, the setup of randomly permuted dataset rows and columns in the toy datasets was one of the important tasks in these works.
- The explanation of how the Wasserstein distance can serve as a tree distance in Proposition 2.1 is unclear. It’s also not evident whether there exists a tree for which the tree distance would correspond to a Wasserstein distance.
- The real-world application is restricted to single-cell RNA sequencing data, despite the introduction citing various data types as motivation.
- The experiment section includes only two competing methods without considering the baselines used in WSV or other distance metric learning approaches.
- $n$ is not yet defined in the Abstract. It’s unclear what does it represent here. In addition, the computational complexity for WSV reported by the author differs from that presented in the original WSV paper.
- In Section 1.1, the authors mention the sliced Wasserstein distance. However, it’s unclear what it is here and how SWD is a special case of TWD. Also, it’s unclear why SWD is not considered an alternative for efficient computation for Wasserstein distance in the WSV framework.
- It’s unclear $\mathbf{w}$ and $\mathbf{Z}$ in line 109 are vector or matrix. Additionally, it’s unclear what is the connection between $\mathbf{x}$, $\mathbf{y}$, $\mu$, and $\nu$ in line 111.
- The notation of Wasserstein distance $\\mathcal{W}\_C$ in line 92 and the notation of TWD $\\mathcal{W}\_\\mathcal{T}$ in line 111 are confusing. In the formal $C$ is a pairwise distance matrix, and in the later $\\mathcal{T}$ is a tree.
- It’s unclear what are $\mathbf{a}$ and $\mathbf{b}$ in Eq.(2). In addition, it’s unclear what size($\mathbf{w}$) represents in line 116.
- More details are needed for how “TWD is a good approximation of the full 1-Wasserstein distance”. What do the authors refer to as “approximation”? What is the relation between TWD and full 1-Wasserstein distance?
- It’s unclear what is $R$ in line 151.
- The authors keep using the term “basis” throughout the paper, e.g.,  tree parameter basis set, the set of basis vector, matrix’s basis set. However, it’s unclear what does it represent in these contexts.
- It’s unclear what "full WSV" represents. Is it different from WSV?
- It’s unclear why the size of the vectors of edge weights is less than the number of nodes in the tree in Section 2.1.
- It’s unclear what $\mathcal{W}_A$ and $\mathcal{W}_B$ represent in Proposition 2.1. Here, $A$ and $B$ are the sets, which are not pairwise distance matrix nor tree as in previous notations.
- It’s unclear what $\circ$ denote in Proposition 2.1. Also, it’s unclear what are $\\lambda\_A$, $\\mathbf{z\_i}\^{(\\mathbf{A})}$, and $\\mathbf{Z}\^{\\mathbf{B}}$.
- The proof for Proposition 2.1 in Appendix A is very hard to follow. The notations used are not consistent with those used in the main texts. The newly defined notation is very dense. Also, it’s unclear what are $\\mathbf{W}\_{\\mathbf{A}}$, $\\Phi\_{\\mathbf{S}}$
- More details and explanations are needed for how Theorem 2.2 supports unique and non-zero solutions in Proposition 2.1.
- Algorithm 1 is very hard to follow. For example, it’s unclear what the line “$\\mathbf{Z}\_{\\mathbf{diff}}$ …… “ represents. It’s unclear what are $\\mathbf{A}\_{leaf}$, $\\mathbf{w}\_{\\mathbf{B}}$(prev)
- The reference style is inconsistent: some entries lack publisher information, some links are not official paper links, and "Wasserstein" is sometimes written with a lowercase "w."
- The notation style is inconsistent: vectors and matrices are inconsistently represented, with a mix of boldface and regular type. The notation for the tree parameter in Section 1.1 is different than in Section 2.1

## Minor
- Missing “-” for tree-Wasserstein distances in line 077 and line 102
- The acronym "OT" is used without being defined first
- Missing punctuations in equations
- It’s unclear what is $TB$ in line 214

[1] Ankenman, J.I., 2014. Geometry and analysis of dual networks on questionnaires. Yale University.

[2] Mishne, G., Talmon, R., Cohen, I., Coifman, R. R., & Kluger, Y. (2017). Data-driven tree transforms and metrics. IEEE transactions on signal and information processing over networks, 4(3), 451-466.

[3] Gavish, M. and Coifman, R.R., 2012. Sampling, denoising and compression of matrices by coherent matrix organization. Applied and Computational Harmonic Analysis, 33(3), pp.354-369.

[4] Shahid, N., Perraudin, N., Kalofolias, V., Puy, G. and Vandergheynst, P., 2016. Fast robust PCA on graphs. IEEE Journal of Selected Topics in Signal Processing, 10(4), pp.740-756.

[5] Yair, O., Talmon, R., Coifman, R.R. and Kevrekidis, I.G., 2017. Reconstruction of normal forms by learning informed observation geometries from data. Proceedings of the National Academy of Sciences, 114(38), pp.E7865-E7874.

**Questions:**

- In line 194, what does “$a_i$ defined as for WSV” mean?
- What is the size of $\mathbf{U}$ in line 259?
- How does the choice of tree construction affect the proposed method? If a different tree construction method were used instead of ClusterTree, would it impact the method's outcome?
- How to decide whether the algorithm reaches convergence in Algorithm 1?
- How fast does the proposed algorithm converge? How does the performance change across the iteration?
- What are the hyperparameters of the proposed method? How sensitive are they in the experiments?
- What is the Euclidean metric baseline in Table 1?
- Why is WSV not considered as a baseline in Table 1?
- Why are other TWD methods not considered baselines in Table 1?

---

> ### Author Response · Authors · 2024-11-21
> **Reviewer 1: Improved submission addressing notation, existence of a solution, convergence, and all specific points**
>
> (Part 1) We thank the reviewer for a thorough read and suggestions. We appreciated the comment that the "proposed method demonstrates efficiency compared to WSV and SSV presented in Huizing et al. (2022)" and we have addressed the following major points in our reply to all reviewers above:
> * Notation
> * Proving there is a solution to the power iterations problem
> * Convergence (empirically)
>
> In terms of the specific weaknesses (we answer in the same order to help the reviewer to cross-reference):
>
> * We were glad to read the related works mentioned, and apologise for not being aware of these -- [2] (Mishne et al.) is particularly relevant for us (and could inspire possible extensions from our work!). We also thought [1] (Ankenman)'s construction of partition trees using a random walk algorithm was intriguing. We have now included a paragraph about these works starting at line 170. Specifically, we write:
>
> _While our approach to improve complexity through embedding samples and features as leaves on trees was inspired from the TWD literature, several authors have explored the relationship between samples (rows) and features (columns) as distributions over the other in general (Ankenman, 2014; Gavish & Coifman, 2012), including on graphs (Shahid et al., 2016) and in tree-embeddings (Anken- man, 2014; Mishne et al., 2018; Yair et al., 2017). These methods do not learn the ground metric in the same way as our proposal, but Mishne et al. (2018), Ankenman (2014) and Yair et al. (2017) describe iterative metric-learning of the tree metric and tree construction, which is related._
>
> * Regarding lack of clarity on the explanation of "how the Wasserstein distance can serve as a tree distance in Proposition 2.1", we have now rephrased the explanation starting at line 231 to explain the intuition in the embedding, and been more precise about the nature of the approximation of TWD to 1-WD.
>
> * On ``It’s also not evident whether there exists a tree for which the tree distance would correspond to a Wasserstein distance'' please see Appendix A where Proposition 2.1 is proved and refer to Yamada et al., 2022 [3]: Proposition 1 in Yamada et al. ensures that for any distance metric on the support of two discrete measures, there exists a tree such that TWD equals to the 1-Wasserstein distance with that metric (using the shortest path on the tree for the tree metric).
>
> * We agree that the real-world application was restricted to single-cell RNA sequencing data, while the introduction cites various data types as motivation. We have run the algorithm on orientation tuning in calcium-imaged neurons in V1 with interpretable results (we get sharper ring-like structures as compared to Euclidean metric, but these are still visible albeit with more noise using just a Euclidean metric, as might be expected for an orientation dataset). We could include these results as an appendix, but we wonder whether it would be best to remove the references to neural data in the introduction and leave these as options for future work, given the space limitations (which is what we have done for now). Do you have any preference as a reader to either of these two options?
>
> * Regarding only comparing the two competing methods: we include a full table of comparisons for one of the datasets in the Appendix, which showcases other methods' poorer performance. If it would assure the reviewer, we can run a subset of these on the other datasets, too. On the second point, since our main aim was to show that the Tree-WSV algorithm improves on WSV in terms of compute time and outperforms SSV in terms of accuracy, we did not include other distance metric learning approaches. If the reviewer has any in particular to recommend, we are happy to compare these.
>
> * In terms of complexity on $n$: We realise this was unclear in the abstract vs. later in the paper, and thank you for pointing it out. We assumed that the original $n,m$ in the Huizing paper were of a similar order and dropped the log term to get to an *approximate* quintic complexity. However, this is indeed unclear and not explained, so we will instead use the full complexity: $\mathcal{O}(n^2m^2(n log(n) + m log(m))$ for $n$ samples and $m$ features (l.018 in updated manuscript).
>
> (See next part)
>
> [1] Ankenman, J.I., 2014. Geometry and analysis of dual networks on questionnaires. Yale University.
>
> [2] Mishne, G., Talmon, R., Cohen, I., Coifman, R. R., & Kluger, Y. (2017). Data-driven tree transforms and metrics. IEEE transactions on signal and information processing over networks, 4(3), 451-466.
>
> [3] M. Yamada, Y. Takezawa, R. Sato, H. Bao, Z. Kozareva, and S. Ravi. Approximating 1-Wasserstein
> distance with trees. Transactions on Machine Learning Research, 2022. ISSN 2835-8856. URL
> https://openreview.net/forum?id=Ig82l87ZVU.

---

> ### Author Response · Authors · 2024-11-21
> **Cont: specific weaknesses (notation, approximation of TWD to 1-WD, extra theory)**
>
> (Part 2)
>
> * We have updated the description of the sliced-Wasserstein distance. To answer the reviewer's question: SWD is a special case of TWD when the tree structure has just a root connected to leaves (which we now write). For this reason, we did not implement SWD as a comparison in this work: there is a loss of expressivity since there are fewer edge weights, so it is a worse low-rank approximation than a larger tree.
>
> * We have updated all notation to match ICLR standards: $\boldsymbol{w}, \boldsymbol{Z}$ are a vector and a matrix respectively (we now state "Let $\boldsymbol{w}$ be the vector of weights"). We also rewrote l. 111 to explain the notation in more detail: "Given a tree metric $d_\mathcal{T}$ on leaves $x,y$ and transport plan $\pi$ between measures $\mu,\nu$, the TWD can be written as (previous expression, does not render here), where $U (\mu, \nu ) =$ (full expression inserted in-text, does not render here) is the set of joint probability distributions with marginals $\mu$ on $\mathcal{X}$ and $\nu$ on $\mathcal{Y}$ ."
>
> * In terms of the notation of Wasserstein distance
> in line 92 and the notation of TWD in line 111: we have used former authors' notation, but we agree it is confusing. We hope that the subscript $\mathcal{T}$ makes clear when we refer to TWD rather than the full 1-WD and have now corrected so that this applies where appropriate; is that the case? If not, do you know of alternative notation that would help make this clear?
>
> * Eq. (2): Thanks for pointing this out, as it was quite confusing. We now write $\boldsymbol{a,b}$ as $\boldsymbol{x,y}$, consistent with l.111 above. And size($\boldsymbol{w}$) is now rewritten as dim($\boldsymbol{w}$).
>
> * Regarding the goodness of approximation of TWD to 1-WD: the paper by Yamada et al. (2022) [3] showing TWD approximates the 1-WD does so mostly with empirics, and while the equivalent empiric results are strong, we do not know a convincing proof of the bound of the approximation. There is a constructive proof in that paper which suggests that our Theorem 2.3 might actually help to show this, but we believe it should stand as future work as it would take significant effort. We have tried to clarify in the rewrite of Proposition 2.1 and onwards that our method takes inspiration from Huizing et al. (2022), who use the full 1-WD, but we use TWD. How well TWD approximates 1-WD is well-established empirically but should be proven (out of scope for our work).
>
> * l. 151: We now define $R$ as a norm regulariser (identical to Huizing's description).
>
> * We recognise the frequent and confusing use of the word "basis." These should all mean the same thing, and we have tried to systematise this in the new lemma / proof format for what was previously Theorem 2.2, as well as the first paragraph in section 2.3.2, where we now define $\boldsymbol{U}$ to be this basis set for $\boldsymbol{Y^\prime}$.
>
> * ``Full WSV'' was used to refer to the full WSV approach, versus the speed-ups like Sinkhorn's SV mentioned in Huizing et al. We have replaced all mentions of "full WSV" with "standard WSV", as we agree that this may read more clearly.
>
> * Regarding clarity on the size of edge weights vs number of nodes in the tree: Each node that is not a root has exactly one edge as parent of that node, and this counts all the edges in the tree. The root's edges are all counted by child-nodes of the root, so that means the number of edge weights is one less the number of nodes in the tree.
>
> * We agree that Proposition 2.1 had a typo as noted by another reviewer, too -- we have amended so that $W_{B}$ is now $W_{\mathcal{T}_{B}}$ (and the same for the As), thanks!
>
> * We apologise for the lack of definition of some notation. We have now made sure that $\circ, \lambda_A$ are defined in Prop 2.1 and $\boldsymbol{z_i^{(A)},Z^{(B)}}$ in the paragraph preceding. We have also made the notation in Prop 2.1's proof in the appendix consistent, and defined $\Phi$ in Section 1.1 when we address Huizing et al.'s iterative SV approach, and repeated this in the proof.
>
> * We have rephrased Theorem 2.2. into a lemma and a theorem, a suggestion by another reviewer which we think significantly clears how the theory supports unique, non-zero solutions to prop 2.1. The lemma is identical to previous Theorem 2.2. The Theorem reads (sic): " Given any tree $T_A$ with leaves the rows of a data matrix $X$ such that the root node of $T_A$ has degree 3 or more, and a tree $T_B$ with leaves given by $B$ the columns of $X$, there exists a unique, non-zero solution for $\boldsymbol{w_A}$ in the system of linear equations."
>
> * We also have a new Lemma and proof showing that power iterations also have a solution, which deepens the theoretical underpinnings.

---

> ### Author Response · Authors · 2024-11-21
> **Cont: references, minor points, and answers to questions**
>
> (Part 3)
>
> * We tried to simplify the notation in Algorithm 1, and more importantly now introduce this notation in the text (under section 2.3.2 largely), which we hope will help the reader. We found a typo -- $A_{leaf}$ should be replaced with the number of edges -- and we now define convergence. Thanks for noticing this!
>
> * References have been updates to include publisher, links to official papers, and capital letters where needed. We apologise for the rushed impression of the referencing section at the time of initial submission.
>
> * Notation has all been updated to meet ICLR standards and to be consistent between section 1.1. and 2.1.
>
> * Regarding minor points, all four have been corrected. We thank the reviewer for taking their time to find these!
>
> Questions:
>
> * l. 194: We mean that we use the same notation i.e. $\boldsymbol{a_i }$ is a normalised row in a data matrix. We now simply rewrite this to avoid reader fatigue paging back and forth.
>
> * We have added the size of $\boldsymbol{U}$ in the manuscript: in text, it is number of edges x number of edges.
>
> * We do not believe choice of tree construction influences the algorithm, up to the number of edges in the tree and whether the root has 3 or 2 children. We investigate these properties in Figure 2. We have also tried methods like QuadTree, and get similar results, except of course for differences in the time it takes to construct the tree. Alternative clustering methods to construct trees could be an avenue for future exploration as this could impact the meta-iterations, but learning the tree weights is identical on any tree structure.
>
> * Regarding convergence, we have added our empiric stop-point / threshold to the algorithm for when weight vectors converge, and make a few more comments on convergence in Appendix E, where we also show empirical convergence for different dataset sizes, with dynamics matching those in Huizing et al.
>
> * For hyperparameters, we set the number of children in each ClusterTree and the convergence threshold. The number of children / tree structure is briefly discussed and compared in Figure 2 and surrounds. The convergence threshold does not seem to matter as empirically we observe convergence to small values quickly.
>
> * The Euclidean metric baseline uses the Euclidean metric (i.e. L2 norm) as the distance between features (rather than say our learned metric); we then compute the ASW based on that metric (a bit like k-means distances between samples compared on labelled clusters).
>
> * We do not use WSV as a baseline in Table 1 because it is simply too computationally expensive to even run! It would take an estimated over 30 hours just to run the smallest genomics dataset in our empiric section. Note that even in the original paper by Huizing et al. (2022) they only used SSV on their genomics dataset, which is identical to our first dataset (PBMC) -- we did exactly the same, and got the same ASW score for SSV.
>
> * Regarding the use of other TWD methods as baselines: we are not sure what you mean here. Can you clarify which other TWD methods you are referring to? We are not aware of any that learn ground metrics except our own.
>
> Thanks again for these detailed comments and we look forward to engaging more!

---

> > ### Comment · Reviewer_1BGM · 2024-11-27
> >
> > I thank the authors for their detailed response.
> >
> > Regarding the use of other TWD methods as baselines, I would like to clarify my point. Since TWD is an approximation of the Wasserstein distance on a ground metric by approximating the ground metric with a tree metric, and since the WSV involves computing the Wasserstein distance between samples and using it as the ground metric to compute the Wasserstein distance between features, it seems natural to consider computing the TWD between samples and using it as the ground metric to compute the TWD between features using alternative TWD methods. For example, using cTWD from Yamada et al., 2022, one could compute the cTWD between samples and then use it as the ground metric to compute the cTWD between features, and iterate it. Given that your work is motivated by WSV, incorporating this approach as a baseline seems like a natural extension to evaluate. How does this approach differ from your method?
> >
> > In addition, the description in line 231 and Appendix A did not sufficiently address my concern about “The explanation of how the Wasserstein distance can serve as a tree distance in Proposition 2.1 is unclear. It’s also not evident whether there exists a tree for which the tree distance would correspond to a Wasserstein distance.”
> >
> > First, I’m not sure how the description in lines 704-706 in Appendix A relates to Le et al., 2019 and Yamada et al., 2022. From Proposition 1 in Le et al., 2019 (stated as Proposition 2 in Yamada et al., 2022), it was shown that given a tree $T$ and probability measures supported on $T$, the TWD computing using $T$ is the Wasserstein distance on the tree metric $d\_T$, i.e., when the ground metric is the tree metric.
> > In addition, in your response, you mentioned that "Proposition 1 in Yamada et al., 2022 ensures that for any distance metric on the support of two discrete measures, there exists a tree such that the TWD equals the 1-Wasserstein distance with that metric (using the shortest path on the tree for the tree metric)." However, it is unclear to me how this result extends to Proposition 2.1 in your work, as you consider n histograms for samples (and m histogram for features) rather than just two. It was not clear to me how there exists one same tree such that the pairwise Wasserstein distance can serve as the pairwise tree distances. I am concerned because what you claimed in Proposition 2.1 and Appendix A is very strong, and it implies that tree distance can approximate any distance metric, including Wasserstein distance, without any distortion and I’m not sure this is correct.
> >
> > Could you clarify these points and whether there are limitations or assumptions I may have overlooked?

---

> ### Author Response · Authors · 2024-11-29
> **Addressing part 1: using other TWD baselines**
>
> We thank the reviewer for engaging and providing clarification on their point about alternative TWD methods (the first part of the follow-up). The question is a good one: indeed, the method by Yamada et al. (2022) really did inspire this work and is aligned since TWD is calculated in the same way and we also learn edge weights – so much so that when we explored this idea originally, we did it in the way suggested by the reviewer (using cTWD to compute cTWD between samples and subsequently using this as a ground metric between features, then iterating with the tree construction as per Yamada et al. (2022))! However, we found empirically that the algorithm did not seem to converge over several SV iterations, and reasoned that it could not scale to reasonable $n$ and $m$ greater than 500. This is because of two problems which motivated many of the extensions in our paper: first, cTWD as it is stated required computing all  $n^2$ or $m^2$ tree-Wasserstein distances at each iteration. Second, solving the LASSO problem in the way suggested is computationally expensive and memory-inefficient, because the full $n^2$ or $m^2$ distances data is ideally used as input.
>
> To expand on time complexity:  each LASSO has complexity $\mathcal{O}(p^3+kp^2)$ where $p$ are the number of features and $k$ the number of samples (Efron et al., 2004 [1]). In the case of cTWD, $p=N, k=n^2$ (N is number of edges and n number of leaves), so the total complexity is at least $\mathcal{O}(n^4)$, since $n<N<n^2$. In addition, calculating all $n^2, m^2$ TWDs still has $mn^2$ and $nm^2$ (resp.) complexity. So each iteration is at least quartic in the bigger of $n,m$ in complexity, which is worse than the complexity of SSV. Our solution – to reduce the problem to solving for a subset of the full leaves (with related Lemma 2.3, Theorem 2.4) – moves this to cubic complexity by formalising the set-up in Yamada et al. to a linear system of equations problem. Indeed, one could apply our new theory to the Yamada et al. 2022 tree weight calculation (with non-negative LASSO or NNLS as the solver) and presumably speed up the tree weight calculation there!
>
> To expand on memory consumption: for even relatively small $m$ or $n$, we have to calculate the same $Y^\prime$ matrix for each iteration (shape $(N-1)\times n^2$) and this is prohibitive quickly (even for $n=500$, this becomes a problem!). We solve this by using SVD or the recursive algorithm in the appendix to make $U$, which has size $(N-1)\times(N-1)$.
>
> Why were these not a problem for the Yamada et al. paper which used a single weight-learning iteration to calculate a single tree? First, their use-case is on word-embeddings with a tree of ~5000 leaves (words), where each word has an embedding dimension of 300. Computing the Euclidean distances between words when each is 300-dimensional is fast and can be done in parallel (we also use parallel processing, albeit via JAX). In addition, while it is relatively quick to calculate a subset of TWDs (hundreds – as they used to compare to full Wasserstein distance), this becomes much worse computationally when computing all $5000^2$ TWDs for all document pairs – our method would require only computing at most 9999 at each iteration. Second, and importantly, Yamada et al. (2022) used a trick with their hyperparameters in the algorithm that sped things up: they set a max number of pairwise distances to sample in the weight-finding algorithm. This effectively reduces the size of the $Y^\prime$ matrix and the least squares (LASSO) problem, although it does so randomly. They account for some of this randomness using a sliced version and averaging across tree constructs. We think this could be an interesting (stochastic version) algorithm to explore, but the randomness seemed to induce errors when we tried to run a modified version of Yamada et al. with singular value iterations over edge-weight learning (both previously and in the last few days). In particular, the algorithm did not appear to converge in a reasonable number of iterations on the toy dataset (using $m,n=200,300$ and hyper-parameter of number of random sub-samples 100 or 1000). From Huizing et al., 2002 [2] Figure 3, we assumed that stochastic iterations in general may have low yield / take longer to converge, but notably we have not tested this thoroughly in the tree setting – we agree that this would be good to follow up.
>
> We hope this explanation as to why we did not include cTWD as a comparative method helps! For these reasons, we chose to focus on the theory towards reducing the size of the least squares problem. However, it would be useful to try to compare stochastic-type approaches with averages across trees (or similar) as an alternative in the future, and we would be happy to add this as a suggestion.
>
> [1] Efron et al., Annals of Statistics, 2004 (https://arxiv.org/pdf/math/0406456)
> [2] Huizing et al., ICML, 2022 (https://proceedings.mlr.press/v162/huizing22a/huizing22a.pdf)

---

> > ### Author Response · Authors · 2024-11-29
> > **Addressing part 2: proposition 2.1 and the approximation of TWD to full Wasserstein distance**
> >
> > We again thank the reviewer for this insightful comment. We apologise for not making this point clearer in-text: it was also picked up by reviewer 2. Importantly, we do not wish to assert that our method using TWD approximates well the Wasserstein distance theoretically such that _any_ distance could be learned, and even though we and Yamada et al. 2022 observed good approximation (vs WSV) empirically, we agree that there could be some distortion (Tree-WSV is a low-rank approximation by definition). Hence, Prop 2.1 only refers to the singular vector problem on the two trees (i.e. ground metric on one induces TWD on the other, as opposed to on the general Wasserstein distance), and we should be more careful in the statements in the appendix.
> >
> > What this means is that lines 704-706 in the appendix do not affect the proof of the singular vector proposition on trees, but we agree that these lines should be changed or removed: indeed, we overstated the Propositions in Le et al. 2019 and Yamada et al. 2022, as while it is true that a tree exists on which TWD approximates the Wasserstein distance, we are not guaranteed to be able to construct it (as we imply from Prop 2 of Yamada et al.). The way you have stated these lines is indeed much clearer ("it was shown that given a tree $T$ and probability measures supported on $T$, the TWD computed using $T$ is the Wasserstein distance on the tree metric $d_T$, i.e., when the ground metric is the tree metric"). On your second point -- this is also a good catch: we could not adopt a proof to show that a tree exists supporting $n$ samples (as opposed to just 2, which is what the theory states, as you rightly point out). This actually suggests an improvement to the algorithm that we will add in future work: we could learn several trees (sliced version) and average to try get a better approximation. We are sorry for this confusion, and thank you for pointing it out!
> >
> > To make it clearer, would it help to restate l. 704-706? Perhaps: "From Yamada et al. and Le et al., given a tree $T$ and two probability measures supported on $T$, the TWD computed using $T$ is the 1-Wasserstein distance on the tree metric $d_T$. There also exists a tree such that for any two measures, it can approximate the 1-Wasserstein distance between the measures; however, since this is only proved for two measures, combining many measures (samples) as we do in this work is not guaranteed to represent the Wasserstein distance (or indeed any distance) without distortion. In this proof we seek to show that we can construct a TWD-version of the Wasserstein singular vector problem, but we do not comment on how well this approximates the 1-Wasserstein distance theoretically." We also believe we should alter the following lines in the proof: "Wasserstein $-->$ tree Wasserstein" (l. 716) and remove "can approximate the 1-Wasserstein distance" in the following line. The calculations that follow stand correct to our knowledge, but notably do not assert that the distance can be learned, and only set-up the problem. We need the theory in Lemma 2.2 to show that a solution to the singular vector problem exists, but indeed the solution is really just to the set of equations we write, and therefore could have distortion as compared to the 1-WD (since pairs of TWDs are not guaranteed to represent the 1-WD).

---

> > > ### Comment · Reviewer_1BGM · 2024-12-01
> > >
> > > Thank you for the detailed clarification and for addressing these points.
> > >
> > > Your remarks on these points are much clearer and could improve the clarity and accuracy of the manuscript in terms of the empirical problem set-up and the theoretical insights. I believe incorporating these notes/remarks and revising the corresponding texts as proposed would certainly make the distinction clearer and mitigate any potential confusion.
> > >
> > > I thank the author again for their effort to address my comments and the work they put into the rebuttal and discussion. I am satisfied with the proposed adjustments and look forward to seeing the improved version.

---

> > > > ### Author Response · Authors · 2024-12-01
> > > > **Thanks; and we will incorporate these comments in the revised manuscript**
> > > >
> > > > Thank you for the helpful questions and discussion. We agree, and will definitely incorporate remarks on both the approximation to WSV and the comparison to cTWD in the manuscript (regardless of the ICLR decision -- since we are not able to make revisions in the immediate period). Your thoughts have really been useful in clarifying the work.

---

### Author Response · Authors · 2024-11-21
**Improved submission addressing notation, computational complexity, more rigorous theory regarding existence, convergence, and the ClusterTree algorithm**

We thank all the reviewers for their time and for their clear and thorough comments. The reviews were all insightful and provided helpful areas for development. We believe the updated manuscript addresses the major points raised, and is significantly improved as compared to the original. In particular:

* We have carefully reviewed notation and typos. Notation is now consistent with ICLR standards as well as throughout the paper and appendices. We have sought to simplify notation where it was confusing.

* We have been cautious in directly computing and quoting computational complexity, rather than using loose approximations.

* Thanks especially to an insight from Reviewer 2, we have included a more rigourous approach to the theory. In particular:
(1) We now write what was previously Theorem 2.2 as a Lemma and use this to prove that each least squares iteration has a unique solution in the subsequent Theorem.
(2) We also have carefully re-read Huizing et al., and used elements from there to show that power iterations also must have a solution.

* We now provide some more insight on convergence, including empirics.

* We have clarified and added details on the ClusterTree algorithm used in the paper.

To each reviewer, we will also address specific points in direct replies. Please let us know of any more suggestions, as we found the dialogue so far incredibly useful in honing this work!

---

### Meta-Review · Area_Chair_HpRK · 2024-12-17

**Metareview:**

The paper introduces a computationally efficient method for unsupervised ground metric learning using the tree-Wasserstein distance as a low-rank approximation of Wasserstein singular vectors. The key points raised during the reviewer discussion included concerns about theoretical guarantees, computational complexity, and real-world applications. Specifically, reviewers identified unclear explanations of the relationship between tree distances and Wasserstein metrics, as well as requests for comparisons with additional baselines and further theoretical justification. The authors addressed these concerns by clarifying the notation, improving the theoretical analysis (e.g., existence and convergence of solutions), and adding empirical results for alternative scenarios. These revisions significantly strengthened the paper’s presentation and rigor, leading to the decision to recommend its acceptance as a poster.

**Additional Comments On Reviewer Discussion:**

During the reviewer discussion, key points included the theoretical justification for tree-Wasserstein distance (TWD) approximating the Wasserstein distance, clarification of the ClusterTree algorithm, and comparisons with alternative baselines. Reviewers also raised concerns about notation inconsistencies, computational complexity, and the real-world applicability of the method. The authors addressed these by clarifying theoretical claims, revising notation for consistency, providing additional empirical comparisons, and offering detailed responses on computational scalability. The thorough rebuttal and improvements resolved most concerns, demonstrating the method’s robustness and contributions, which were weighed positively in the final decision to recommend acceptance.

---

### Decision · Program_Chairs · 2025-01-22

Accept (Poster)